# Maintaining Structural Integrity in Parameter Spaces for Parameter Efficient Fine-tuning

**Chongjie Si**[1*], **Xuehui Wang**[1*], **Xue Yang**[2], **Zhengqin Xu**[1], **Qingyun Li**[2],
**Jifeng Dai**[2], **Yu Qiao**[2], **Xiaokang Yang**[1], **Wei Shen**[1†]
[1] MoE Key Lab of Artificial Intelligence, AI Institute, Shanghai Jiao Tong University
[2] OpenGVLab, Shanghai AI Laboratory
{chongjiesi, wei.shen}@sjtu.edu.cn

## Abstract

Adapting pre-trained foundation models for various downstream tasks has been prevalent in artificial intelligence. Due to the vast number of tasks and high costs, adjusting all parameters becomes unfeasible. To mitigate this, several fine-tuning techniques have been developed to update the pre-trained model weights in a more resource-efficient manner, such as through low-rank adjustments. Yet, almost all of these methods focus on linear weights, neglecting the intricacies of parameter spaces in higher dimensions like 4D. Alternatively, some methods can be adapted for high-dimensional parameter space by compressing changes in the original space into two dimensions and then employing low-rank matrix adaptations. However, these approaches destructs the structural integrity of the involved high-dimensional spaces. To tackle the diversity of dimensional spaces across different foundation models and provide a more precise representation of the changes within these spaces, this paper introduces a generalized parameter-efficient fine-tuning framework, designed for various dimensional parameter space. Specifically, our method asserts that changes in each dimensional parameter space are based on a low-rank core space which maintains the consistent topological structure with the original space. It then models the changes through this core space alongside corresponding weights to reconstruct alterations in the original space. It effectively preserves the structural integrity of the change of original N-dimensional parameter space, meanwhile models it via low-rank tensor adaptation. Extensive experiments on computer vision, natural language processing and multi-modal tasks validate the effectiveness of our method.

## 1 Introduction

The recent introduction of foundation models Brown et al. (2020); Kirillov et al. (2023); Devlin et al. (2018); Liu et al. (2019) has demonstrated unparalleled performance and potential across diverse domains in artificial intelligence. Traditionally, the adaptation of pre-trained models for downstream tasks is achieved through fully fine-tuning of all parameters Ma et al. (2024); Raffel et al. (2020); Qiu et al. (2020). However, as the parameter count of these foundation models escalates, the conventional approach to fully fine-tuning becomes prohibitively expensive in various aspects.

To tackle this challenge, recent works Chen et al. (2024a); Guo et al. (2020); He et al. (2021a); Hu et al. (2021a) have focused on the concept of parameter-efficient fine-tuning (PEFT), aiming to minimize the number of adjustable parameters required while achieving optimal task performance. These works specifically explore methods to model the incremental update of pre-trained weights in a manner that is economical in terms of parameters, without necessitating changes to the model's architecture Zaken et al. (2021); Guo et al. (2020); Hu et al. (2021a). Among these works, LoRA Hu et al. (2021a) emerges as a pioneering effort, which proposes to adopt an additional term with a low-rank structure to the original weight. Specifically, the original weight matrix $\mathbf{W}_0 \in \mathbb{R}^{d_1 \times d_2}$

---

*Equal contributions.
†Corresponding author.

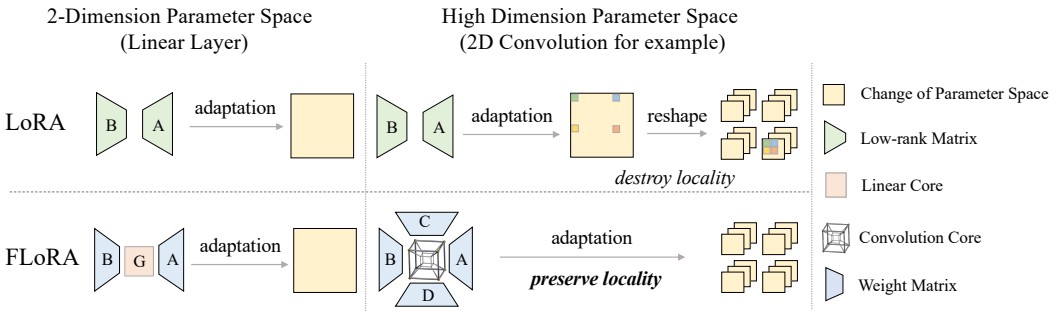

Figure 1: Difference between LoRA and FLoRA. LoRA employs low-rank matrix adaptation for each dimensional parameter space. However, for parameter space of convolution layer, the reshaping operation required by LoRA causes adjacent elements within the kernel to be separated in the matrix, disrupting the spatial locality inherent in the original convolutional space. Conversely, FLoRA asserts that the alternations of each dimensional parameter space has a low-rank core space with the consistent topological structure. This framework enables FLoRA to effectively preserve the structural integrity of the original parameter space, such as maintaining the spatial locality in convolutional operations.

remains frozen, while a learnable low-rank $\Delta\mathbf{W}$ is added to $\mathbf{W}_0$, with the form as

$$\mathbf{W}_0 \rightarrow \mathbf{W}_0 + \Delta\mathbf{W} = \mathbf{W}_0 + \mathbf{B}\mathbf{A}^{\mathsf{T}}, \tag{1}$$

where $\mathbf{A} \in \mathbb{R}^{d_2 \times r}$ and $\mathbf{B} \in \mathbb{R}^{d_1 \times r}$ with $r \ll \{d_1, d_2\}$. Given that $r$ is significantly smaller than the dimension of $\mathbf{W}_0$, LoRA necessitates the updating of only a limited number of trainable parameters, while achieving comparable performances. Subsequent to LoRA, recent studies He et al. (2021a); Bershatsky et al. (2024); Zhang et al. (2022a) endeavor to explore more efficient low-rank matrix adaptation methods concerning the matrix $\Delta\mathbf{W}$.

However, we have observed an intriguing phenomenon: a significant portion of existing works is narrowly focused on two-dimensional parameter spaces (i.e., linear layers), neglecting the existence of other dimensional spaces such as 2D convolution (4-dimension) layers, etc. But in practice, not all layers are linear across various models for downstream tasks. For example, ConvNeXt Liu et al. (2022) and Stable Diffusion Carreira & Zisserman (2017) are two models predominantly utilizing 2D convolutional layers. Alternatively, some approaches can be adapted for high-dimensional parameter spaces by directly adopting low-rank matrix adaptation. They models alternations in high-dimensional spaces into two dimensions, while neglecting the structural complexities of the original spaces. For instance, as detailed in Sec. 2.1 and Fig. 1, LoRA Hu et al. (2021a) compresses the changes in convolutions, which is a four-dimensional parameter space, into two dimensions. It subsequently applies low-rank matrix adaptation to the two-dimension space, intending to represent the alterations of the original four-dimensional parameter space. Yet, as discussed in Sec. 2.2, this adaptation fails to capture the inherent complexity and spatial locality specific to convolution operations. The result is a reshaped two-dimensional structure that compromises the integrity of the original parameter space, leading to a representation that does not fully encapsulate the change of convolutional space.

To this end, in this paper, we propose a low-rank tensor adaptation based method, FLoRA, represented as Fundamental LOw-Rank Adaptation. Positioned as a superior alternative to LoRA, FLoRA meets the ensuing three properties:

- It can identify an appropriate low-rank representation for the changes in various dimensional parameter spaces, without destructing the structural integrity of the original parameter spaces.
- It can maintain a consistent formulation across various dimensional parameter spaces.
- When applied to linear weights, with the same parameter budget, it requires similar training time and resources to that of LoRA, yet yield superior performance.

Specifically, since a much lower rank than the direct rank of parameter space is sufficient to represent the original space Aghajanyan et al. (2020); Li et al. (2018), FLoRA asserts that **the alternations of**

**each dimensional parameter space, whether 2D or 4D, have a corresponding core space. This core space is low-rank and retains the same spatial dimensions (i.e., 2D or 4D) as the original parameter space, suggesting that they share a consistent topological structure**. FLoRA then models the alternations by using this core space combined with a series of weights to reconstruct alternations in the original parameter space. Thanks to the intrinsic properties in the structure of the core space, FLoRA efficiently maintains the structural integrity of the original parameter space. Extensive experiments are conducted with several pretrained models on computer vision, natural language processing and multi-modal tasks, which validates that regardless of the model, the kind of downstream task, or the dimensionality of the parameter space, FLoRA's performance surpasses that of LoRA and other existing methods.

The contributions of this paper are as follows:

- We propose a novel PEFT method, FLoRA. To the best of our knowledge, it is the first time that a PEFT method has been designed for different dimensional parameter spaces, aiming to preserve their topological structure while seeking low-rank representations.
- Extensive experiments on different tasks, include computer vision, natural language processing and multi-modal tasks, demonstrates that FLoRA significantly surpasses other baselines, validating the effectiveness of FLoRA.

## 2 PRELIMINARIES

### 2.1 LOW RANK ADAPTATION

LoRA Hu et al. (2021a) models the incremental update of a pre-trained weight matrix $\mathbf{W}_0 \in \mathbb{R}^{d_1 \times d_2}$ by the product of two small matrices $\mathbf{A} \in \mathbb{R}^{d_2 \times r}$ and $\mathbf{B} \in \mathbb{R}^{d_1 \times r}$, where $r \ll \{d_1, d_2\}$. For $\mathbf{h} = \mathbf{W}_0\mathbf{x}$, the modified forward pass is

$$\mathbf{h} = \mathbf{W}_0\mathbf{x} + \Delta\mathbf{W}\mathbf{x} = \mathbf{W}_0\mathbf{x} + \mathbf{B}\mathbf{A}^\mathsf{T}\mathbf{x}. \tag{2}$$

Matrix $\mathbf{A}$ is initialized with a random Gaussian distribution, and $\mathbf{B}$ with zeros, setting the initial $\Delta\mathbf{W}$ to zero for training. LoRA's application is straightforward to the linear layers, while for a convolution layer characterized by weights $\mathcal{W}_c \in \mathbf{R}^{d_{in} \times d_{out} \times k \times k}$, with $d_{in}$ / $d_{out}$ denoting the dimension of input / output respectively and $k$ representing the kernel size, LoRA is adapted based on matrix decomposition:

$$\mathcal{W} = \mathcal{W}_c + \Delta\mathcal{W} = \mathcal{W}_c + Reshape(\mathbf{B}_c\mathbf{A}_c^\mathsf{T}, \mathcal{W}_c), \tag{3}$$

where $\mathbf{A}_c \in \mathbb{R}^{kd_{out} \times r}$ and $\mathbf{B}_c \in \mathbb{R}^{kd_{in} \times r}$. Here, $Reshape(\mathbf{B}_c\mathbf{A}_c^\mathsf{T}, \mathcal{W}_c)$ involves altering the dimensions of $\mathbf{B}_c\mathbf{A}_c^\mathsf{T}$ to match those of $\mathcal{W}_c$. It is obvious that LoRA unfolds the original 4-dimensional parameter space $\Delta\mathcal{W}$ into a 2-dimensional space, subsequently leveraging the low-rank approximation of this 2-dimensional space to represent the original 4-dimensional construct. As shown later, this low-rank matrix adaptation method will destruct the structural integrity of the convolution layer.

### 2.2 WHY MATRIX ADAPTATION BREAKS THE STRUCTURAL INTEGRITY OF THE CONVOLUTION?

The changes of a high-dimension tensor such as convolutional parameter space can be modeled based on low-rank matrix adaptation following Eq. (3). However, during the reshaping process, elements that are adjacent within the convolutional kernel are those dispersed across various positions in the original matrix. More specifically, elements that are now localized within the convolutional kernel were located across multiple rows or columns of the matrix for reshaping. This shift poses significant challenges in learning spatial correlations among elements positioned disparately. Therefore, such a transformation disrupts the principle of locality inherent in the original convolutional operation, where each output element is determined by a small region of the input.

## 3 METHOD

In this section, we first introduce the formulation of FLoRA in an N-dimensional parameter space. Specifically, for a pre-trained weight $\mathcal{W}_0 \in \mathbb{R}^{I_1 \times I_2 \times \cdots \times I_N}$ with N-dimension, we update the weights

$\mathcal{W}_0$ with the changes $\Delta\mathcal{W}$ as

$$\mathcal{W}_0 \to \mathcal{W}_0 + s * \Delta\mathcal{W} = \mathcal{W}_0 + s * \mathcal{G} \times_1 \mathbf{A}^{(1)} \times_2 \mathbf{A}^{(2)} \times \cdots \times_N \mathbf{A}^{(N)} \tag{4}$$

without loss of generality. Here $\mathcal{G} \in \mathbb{R}^{J_1 \times J_2 \times \cdots \times J_N}$ and $\mathbf{A}^{(n)} \in \mathbb{R}^{I_n \times J_n}$, where $J_n \ll I_n$. $J_1, ..., J_N$ are hyper-parameters, and $\times_n$ denotes the mode-$n$ product between a tensor and a matrix. FLoRA considers the tensor $\mathcal{G}$ as a low-rank core space with the consistent topological structure to the original parameter space, with $\mathbf{A}^{(n)}$ representing the weights associated with each dimension. We then scale $\Delta\mathcal{W}$ by $s$ with $s$ being a constant. In the subsequent subsections, we will detail its manifestations in convolution and linear layers.

## 3.1 FLoRA for Convolution Layer

Convolution operations in deep learning are characterized by a four-dimensional parameter space, encapsulated in a weight tensor $\mathcal{W}_c \in \mathbf{R}^{d_{in} \times d_{out} \times k \times k}$, with $d_{in}$ / $d_{out}$ denoting the dimension of input / output respectively and $k$ representing the kernel size. A pivotal property to consider within $\mathcal{W}_c$ is the spatial locality, which plays an essential role in the convolution layer's ability to compile and process information across the input matrix. This process is facilitated by the kernel's spatial dimensions ($k \times k$), which determines the scope of the input data scrutinized by each convolution operation. To preserve the attributes of the spatial locality and uphold the convolution parameter space's integrity, FLoRA models the update $\Delta\mathcal{W}$ for a convolution layer as

$$\Delta\mathcal{W} = \mathcal{G} \times_1 \mathbf{A} \times_2 \mathbf{B} \times_3 \mathbf{C} \times_4 \mathbf{D}, \tag{5}$$

where $\mathcal{G} \in \mathbb{R}^{r_1 \times r_2 \times r_3 \times r_3}$, $\mathbf{A} \in \mathbb{R}^{d_{in} \times r_1}$, $\mathbf{B} \in \mathbb{R}^{d_{out} \times r_2}$ and $\mathbf{C}, \mathbf{D} \in \mathbb{R}^{k \times r_3}$. $r_1$ and $r_2$ are two hyper-parameters which are significantly smaller than $\{d_{in}, d_{out}\}$. $r_3$ is another hyper-parameter that smaller than the kernel size $k$ of a convolution layer. Given that $3 \times 3$ convolution is a prevalent configuration in convolutional foundation models Woo et al. (2023); Rombach et al. (2022); Wang et al. (2023a), $r_3$ is consequently chosen from $\{1,2\}$.

The core tensor $\mathcal{G}$ in FLoRA can be viewed as a compressed convolution parameter space. In essence, it serves as a core space for convolution. This means that in any convolution layer, there exists a convolution core, and what FLoRA aims to do is to determine this convolution core for each convolution space and configure the corresponding weights $\mathbf{A}$, $\mathbf{B}$, $\mathbf{C}$ and $\mathbf{D}$ for reconstructing that space. Different to low-rank matrix adaptation based methods, FLoRA does not need to alter the structure of the convolution. Alternatively, by learning the convolution core, FLoRA effectively preserves the convolution's property of spatial locality.

Furthermore, while preserving or potentially augmenting the representational power of the convolution process, FLoRA achieves a remarkable reduction in the number of trainable parameters in comparison to LoRA. Assuming that the rank for both the input and output dimensions is uniform ($r_1 = r_2 = r$), the parameter requirement for FLoRA is calculated as $O(r_3^2 r^2 + r(d_{in} + d_{out}) + 2r_3 k)$ parameters, while LoRA necessitate training at least $O(rk(d_{in} + d_{out}))$ parameters. Given that typically $r, k \ll \{d_{in}, d_{out}\}$, therefore, FLoRA exhibits better parameter efficacy than LoRA as the number of the kernel increases.

## 3.2 FLoRA for Linear Layer

For a linear layer with weight $\mathbf{W}_0 \in \mathbb{R}^{d_1 \times d_2}$, FLoRA models the the update $\Delta\mathbf{W}$ as

$$\Delta\mathbf{W} = \mathcal{G} \times_1 \mathbf{A} \times_2 \mathbf{B} = \mathbf{A}\mathbf{G}\mathbf{B}^{\mathsf{T}}, \tag{6}$$

where $\mathbf{G} \in \mathbb{R}^{r_1 \times r_2}$, $\mathbf{A} \in \mathbb{R}^{d_1 \times r_1}$ and $\mathbf{B} \in \mathbb{R}^{d_2 \times r_2}$. Similar to that for convolution layer, the core matrix $\mathbf{G}$ can be viewed as a core space for the 2-dimension parameter space, with $\mathbf{A}$ and $\mathbf{B}$ being the corresponding weights to reconstruct the alternations in linear space.

## 4 Experiment

### 4.1 Models and Datasets

We conduct comprehensive experiments across computer vision (CV), natural language processing (NLP) and multi-modal (MM) tasks.

Specifically, for CV tasks, we employ FLoRA to fine-tune the ConvNeXt-V2-L Woo et al. (2023), evaluating it on MS COCO Lin et al. (2014) by using Mask R-CNN He et al. (2017) implemented in MMDetection Chen et al. (2019), and on remote sensing image datasets DOTA Xia et al. (2018) with Oriented R-CNN Xie et al. (2021) based on MMRotate Zhou et al. (2022), and on the ADE20K Zhou et al. (2017) dataset thanks to UperNet Xiao et al. (2018) integrated in MMSegmentation Contributors (2020). We also employ FLoRA to fine-tune large-scale vision foundation model, i.e. InternViT-6B Chen et al. (2023), on ADE20K datasets. Detailed hyper-parameter settings can be find in Table 9 in Appendix.

For NLP tasks, we evaluate the DeBERTaV3-base He et al. (2021b) with FLoRA on the General Language Understanding Evaluation (GLUE) Wang et al. (2018) benchmark, which includes two single-sentence classification, three similarity and paraphrase and four natural language inference datasets. More details on GLUE dataset can be found in Table 10 in the Appendix.

For multi-modal tasks, we employ FLoRA to fine-tune LLaVA-1.5-7B Liu et al. (2024a), which is composed of a language model, Vicuna-1.5-7B Peng et al. (2023) and a vision encoder, CLIP ViT-L/336px Radford et al. (2021), on visual instruction tuning tasks, which include seven vision-language benchmarks: $VQA^{v2}$ Goyal et al. (2017), GQA Hudson & Manning (2019), VisWiz Gurari et al. (2018), SQA Lu et al. (2022), $VQA^{T}$ Singh et al. (2019), POPE Li et al. (2023), and MMBench Liu et al. (2023).

Moreover, we fine-tune SDXLRombach et al. (2022) with FLoRA for generation task in Appendix A.1, LLaMA3-8B AI@Meta (2024) for commonsense reasoning tasks in Appendix A.2, and ViT-B Dosovitskiy et al. (2021) for VTAB-1K benchmark Zhai et al. (2019) in Appendix A.3.

## 4.2 BASELINES

We compare FLoRA with several state-of-the-art methods: fully fine-tuning, BitFit Zaken et al. (2021), HAdapter Houlsby et al. (2019), PAdapter Pfeiffer et al. (2020), AdaLoRA Zhang et al. (2022a), the most representative low-rank method, LoRA Hu et al. (2021a), and the state-of-the-art low-rank adaption method, DoRA Liu et al. (2024b). Specifically, HAdapter is strategically placed between the self-attention module and the FFN module, and it includes a subsequent residual connection. Conversely, PAdapter introduces a more streamlined design, implementing adapters exclusively after the FFN modules and LayerNorm modules. Furthermore, following Zhang et al. (2022a), we apply AdaLoRA, LoRA and DoRA to all weight matrices or tensors. For more details on the baselines, please refer to their original papers.

## 4.3 IMPLEMENTATION DETAILS

We compare FLoRA with other PEFT methods under different parameter budgets. The hidden dimension of Adapters is chosen from {32, 64}, the budget of AdaLoRA is set as {144, 288, 576} and the rank $r$ of LoRA and DoRA is selected from {2, 4, 8, 16, 32}. Other hyper-parameters are initialized according to their original papers.

Additionally, we simply set $r = r_1 = r_2$ for FLoRA. For CV tasks, when fine-tuning ConvNeXt-V2-L, we set $r_3 = 2$ and $s = 4$. We adjusted FLoRA's $r$ to align its parameter budget closely with

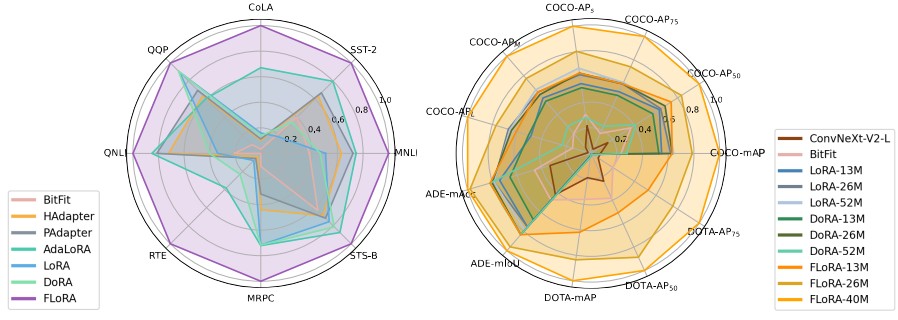

Figure 2: The normalized performance improvement of FLoRA over other baselines

that of other methods for fair comparison. For different methods, we fine-tune all the convolutional layers for ConvNeXt-V2-L (and SDXL). When fine-tuning InternViT-6B, FLoRA's $r$ is set to $\{16, 32\}$, $s = 0.04$, and we fine-tune all the linear layers for different methods. For NLP tasks, FLoRA's $r$ is set to $\{2, 8\}$, and $s = 0.4$. The mean of 5 runs using various random seeds are reported for all the experiments, and all gains have passed the pairwise $t$-test with a significance of 0.05. We fine-tune q, k, v, up and down matrices for all methods. For MM tasks, we set $r = 128$, $s = 1.5$ and fine-tune all linear layers. More training details can be found in Appendix.

We use publicly available PyTorch Paszke et al. (2019) implementation to execute all the baseline comparisons, and all the experiments are conducted on NVIDIA A100 GPUs. The tensor $\mathcal{G}$ is initialized as zero, while other weight matrices are initialized as random Gaussian. For more training details, please refer to the Appendix.

Table 1: Results with ConvNeXt-V2-L Woo et al. (2023) fine-tuned on different datasets. The best performances are shown in bold. "Base" represents for the pre-trained backbone with frozen weights.

| Method | # Params | COCO | | | | | | ADE20K | | DOTA | | | All |
| | | mAP | $AP_{50}$ | $AP_{75}$ | $AP_S$ | $AP_M$ | $AP_L$ | mAcc | mIoU | mAP | $AP_{50}$ | $AP_{75}$ | Avg. |
|---|---|---|---|---|---|---|---|---|---|---|---|---|---|
| Base | - | 37.3 | 63.3 | 39.7 | 27.8 | 41.0 | 46.2 | 59.6 | 48.5 | 31.4 | 61.8 | 27.3 | 44.0 |
| Fully FT | 196M | 52.7 | 74.3 | 58.7 | 38.3 | 56.9 | 67.3 | 64.7 | 53.1 | 33.9 | 59.9 | 34.0 | 54.0 |
| BitFit | 0.2M | 43.1 | 67.6 | 47.4 | 29.5 | 46.7 | 55.3 | 61.2 | 49.1 | 34.6 | 64.2 | 32.9 | 48.3 |
| LoRA | 12.94M | 47.4 | 70.3 | 53.0 | 32.4 | 51.8 | 61.2 | 63.6 | 51.4 | 18.3 | 36.2 | 16.3 | 45.6 |
| DoRA | 13.07M | 47.2 | 69.8 | 52.7 | 32.1 | 51.5 | 61.4 | 63.0 | 50.9 | 19.6 | 37.9 | 17.2 | 45.8 |
| FLoRA | 12.77M | **48.1** | **71.1** | **53.6** | **33.1** | **52.3** | **62.3** | **64.1** | **51.9** | **37.3** | **65.6** | **37.7** | **52.5** |
| LoRA | 25.89M | 48.0 | 70.4 | 53.6 | 33.0 | 52.3 | 62.8 | 63.9 | 51.4 | 20.0 | 38.3 | 18.3 | 46.5 |
| DoRA | 26.16M | 48.1 | 70.7 | 53.6 | 33.1 | 52.1 | 62.6 | 64.0 | 51.9 | 21.1 | 39.7 | 19.1 | 46.9 |
| FLoRA | 25.65M | **49.2** | **71.7** | **54.7** | **34.3** | **53.3** | **63.5** | **65.0** | **52.6** | **38.8** | **68.4** | **39.5** | **53.7** |
| LoRA | 51.78M | 48.2 | 70.7 | 53.7 | 33.4 | 52.5 | 62.7 | 63.9 | 51.6 | 20.4 | 39.4 | 19.1 | 46.9 |
| DoRA | 51.95M | 44.0 | 68.2 | 48.9 | 29.1 | 48.2 | 57.5 | 64.1 | 51.9 | 21.7 | 40.9 | 20.3 | 45.0 |
| FLoRA | 40.49M | **50.4** | **72.6** | **56.2** | **35.4** | **54.6** | **64.8** | **65.1** | **52.8** | **39.7** | **69.0** | **40.9** | **54.7** |

Table 2: Results with InternViT-6B Chen et al. (2023) fine-tuned on ADE20K.

| Method | Base | Fully FT | BitFit | LoRA | DoRA | FLoRA | LoRA | DoRA | FLoRA |
|---|---|---|---|---|---|---|---|---|---|
| # Params (%) | - | 100 | 0.15 | 0.32 | 0.32 | 0.32 | 0.66 | 0.66 | 0.66 |
| mAcc | 64.5 | 70.4 | 68.7 | 69.7 | 69.5 | **70.1** | 70.3 | 69.6 | **71.1** |
| mIoU | 53.0 | 58.2 | 56.1 | 57.1 | 57.3 | **57.7** | 57.3 | 57.1 | **58.0** |
| Avg. | 58.8 | 64.3 | 62.4 | 63.4 | 63.4 | **63.9** | 63.8 | 63.4 | **64.6** |

Table 3: Results with DeBERTaV3-base He et al. (2021b) fine-tuned on GLUE datasets.

| Method | # Params | MNLI m | SST-2 Acc | CoLA Mcc | QQP Acc | QNLI Acc | RTE Acc | MRPC Acc | STS-B Corr | All Avg. |
|---|---|---|---|---|---|---|---|---|---|---|
| Fully FT | 184M | 89.90 | 95.63 | 69.19 | 92.40 | 94.03 | 83.75 | 89.46 | 91.60 | 88.24 |
| BitFit | 0.1M | 89.37 | 94.84 | 66.96 | 88.41 | 92.24 | 78.80 | 87.75 | 91.35 | 86.21 |
| HAdapter | 1.22M | $90.13_{\pm.2}$ | $95.53_{\pm.1}$ | $68.64_{\pm.4}$ | $91.27_{\pm.0}$ | $94.11_{\pm.1}$ | $84.48_{\pm.9}$ | $89.95_{\pm1.1}$ | $91.48_{\pm.3}$ | 88.19 |
| PAdapter | 1.18M | $90.33_{\pm.3}$ | $95.61_{\pm.3}$ | $68.77_{\pm.9}$ | $91.40_{\pm.1}$ | $94.29_{\pm.2}$ | $85.20_{\pm1.2}$ | $89.46_{\pm.7}$ | $91.54_{\pm.7}$ | 88.32 |
| AdaLoRA | 1.33M | $90.38_{\pm.3}$ | $95.87_{\pm.3}$ | $71.45_{\pm1.2}$ | $91.19_{\pm.2}$ | $94.36_{\pm.3}$ | $88.09_{\pm1.9}$ | $90.69_{\pm1.5}$ | $91.84_{\pm.7}$ | 89.23 |
| LoRA | 0.33M | $90.03_{\pm.3}$ | $93.92_{\pm.2}$ | $69.15_{\pm1.2}$ | $90.61_{\pm.1}$ | $93.37_{\pm.3}$ | $85.56_{\pm.7}$ | $90.19_{\pm.7}$ | $90.75_{\pm.2}$ | 87.95 |
| DoRA | 0.41M | $90.21_{\pm.2}$ | $94.38_{\pm.5}$ | $69.33_{\pm1.9}$ | $90.84_{\pm.1}$ | $93.26_{\pm.3}$ | $86.94_{\pm.7}$ | $90.19_{\pm1.2}$ | $91.34_{\pm.9}$ | 88.31 |
| FLoRA | 0.33M | $90.60_{\pm.2}$ | $96.00_{\pm.5}$ | $70.20_{\pm1.1}$ | $91.40_{\pm.4}$ | $94.46_{\pm.6}$ | $88.09_{\pm.5}$ | $90.93_{\pm.6}$ | $91.96_{\pm.7}$ | 89.21 |
| LoRA | 1.33M | $89.80_{\pm.1}$ | $93.69_{\pm.2}$ | $69.30_{\pm1.1}$ | $91.78_{\pm.1}$ | $92.97_{\pm.1}$ | $85.70_{\pm.4}$ | $90.68_{\pm.6}$ | $91.62_{\pm.3}$ | 88.17 |
| DoRA | 1.41M | $89.67_{\pm.2}$ | $94.61_{\pm.5}$ | $69.08_{\pm1.3}$ | $91.80_{\pm.1}$ | $93.23_{\pm.2}$ | $87.33_{\pm1.1}$ | $90.68_{\pm1.0}$ | $91.73_{\pm.5}$ | 88.49 |
| FLoRA | 1.33M | $90.82_{\pm.1}$ | $96.21_{\pm.3}$ | $72.05_{\pm.3}$ | $91.94_{\pm.1}$ | $94.60_{\pm.1}$ | $89.53_{\pm.2}$ | $91.18_{\pm.3}$ | $92.04_{\pm.3}$ | 89.80 |

## 4.4 MAIN RESULTS

We compare FLoRA with other baselines under different parameter budgets. Specifically, ConvNeXt-V2-L is based on convolutions, and other large foundation models are dominantly based on linear layers. We evaluate FLoRA's effectiveness on high-dimension space for the CV tasks, and on linear parameter space for CV and NLP tasks. The results are shown in Tables. 1-4 and Tables. 7-8 in Appendix, and the normalized performance are illustrated in Fig. 2.

For CV tasks, FLoRA employed on ConvNeXt-V2-L achieves superior performances compared to other baselines. On average, FLoRA outperforms LoRA and DoRA by at least **15%** under

Table 4: Results with LLaVA-1.5-7B Liu et al. (2024a) fine-tuned on visual instruction tuning.

| Method | Params (%) | VQA$^{v2}$ | GQA | VisWiz | SQA | VQA$^{\top}$ | POPE | MMBench | Avg. |
|--------|-----------|------------|-----|--------|-----|--------------|------|---------|------|
| Fully FT | 100 | 78.5 | 61.9 | 50.0 | 66.8 | 58.2 | 85.9 | 64.3 | 66.5 |
| LoRA | 4.61 | 79.1 | 62.9 | 47.8 | 68.4 | 58.2 | 86.4 | 66.1 | 66.9 |
| DoRA | 4.63 | 78.6 | 62.9 | 52.2 | 69.9 | 57.0 | 87.2 | 66.1 | 67.6 |
| FLoRA | 4.63 | 78.6 | 62.9 | 51.7 | 70.1 | 57.4 | 87.6 | 66.0 | **67.8** |

different parameter budgets. With nearly **80%** reduction in parameter budget, FLoRA (12.77M) still significantly outperforms LoRA (51.78M), DoRA (51.95M), validating that FLoRA successfully preserves the structural integrity of convolutions. Specifically, FLoRA achieves comparable or even superior performance to fully fine-tuning, while others lag far behind. Additionally, we observe that on tasks with a large domain gap from the pre-training data, e.g. remote sensing images (DOTA), LoRA and DoRA performs significantly worse, while FLoRA maintains consistent superiority. This further validates the robustness of FLoRA, even when facing tasks with large domain gaps. Moreover, when FLoRA is employed to fine-tune InternViT-6B, FLoRA consistently outperforms the baseline across various parameter budgets. Remarkably, by fine-tuning only **0.66%** of the parameters, FLoRA achieves even better performance than fully fine-tuning.

For NLP tasks, FLoRA achieves better or on par performance compared with existing approaches on all datasets under all different parameter budgets. Specifically, under extremely low parameter budget, FLoRA performs better even than the baselines with higher parameter counts. For example, with **0.3M** parameters, FLoRA's performances on SST-2, QNLI, RTE, MRPC and STS-B are all better than baselines with larger parameter budgets.

For multi-modal tasks, FLoRA also achieves SOTA performances compared with baselines. Therefore, at this point, we can conclude that overall, FLoRA achieves remarkable performances across various tasks, model backbones, and dimensions of parameter spaces.

## 5 FURTHER ANALYSIS

In this section, we explore the properties of FLoRA when applied to downstream tasks and in comparison with other baselines like LoRA. We carry out a series of empirical studies to address the following questions: 1) Is the core space truly low-rank? 2) If so, why is FLoRA's low-rank representation better than other methods like LoRA? 3) Does FLoRA require acceptable training costs compared to other methods? 4) Is FLoRA sensitive to the scaling factor? The insights gained from these questions will illuminate the efficacy of FLoRA and guide future research.

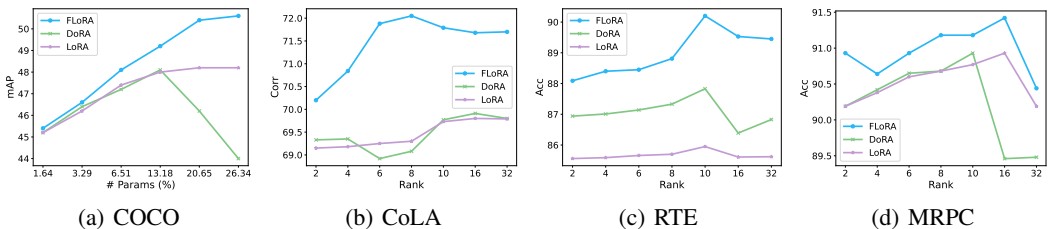

| (a) COCO | (b) CoLA | (c) RTE | (d) MRPC |

Figure 3: Fine-tuning with FLoRA under different rank (parameter budgets).

### 5.1 IS THE CORE SPACE TRULY LOW-RANK?

We present the performance of FLoRA under different rank (i.e., parameter budgets), tested on ConvNeXt-V2-L and DeBERTaV3-base. The results are illustrated in Fig. 3. It is evident that FLoRA's performance with a low rank is comparable to, or even exceeds, that of a higher rank. This finding aligns with observations in Hu et al. (2021a). It indicates that there indeed exists a core space in different dimensional spaces, and the rank of the core space is relatively small. When the rank set is smaller than the rank of the core space, the performance of the model is not optimal. Conversely,

when it exceeds this rank, the core space is completely covered, which introduces some meaningless redundancy and noise. Additionally, we observe that for the convolutional parameter space, the rank of its core space is much larger, for the convolutional space has a more complex topological structure, necessitating a larger rank to adequately describe it.

## 5.2 WHY IS FLORA'S LOW-RANK REPRESENTATION BETTER THAN OTHER METHODS?

Although methods like FLoRA and LoRA can represent changes in the original parameter space using low-rank representations, FLoRA performs better, indicating that its low-rank representation is superior to other forms of low-rank representations. We will explain the reasons as follows. We also provide a comprehensive analysis in Appendix B.1.

In linear parameter spaces, the changes modeled by methods in the LoRA derivatives, can also be simply considered as $\mathbf{AGB}$ with $\mathbf{G}$ being a diagonal matrix. However, FLoRA does not impose any constraints on $\mathbf{G}$, i.e., matrix pattern, which offers several advantages: firstly, it broadens the range of parameter adjustments and enhances the flexibility of parameter learning. Secondly, FLoRA removes the constraint on $\mathbf{G}$. Since any matrix patterns may not represent the characteristics of all downstream tasks, enforcing these constraints on matrix patterns during the learning process could potentially degrade performance. as shown in Tables 1-3. Moreover, Si et al. (2024b) has also theoretically validates the effectiveness of FLoRA, which is significantly easier to reach the global optimum compared to LoRA. This indicates that the low-rank representations of methods like LoRA are inferior to that of FLoRA, thereby yielding inferior performances.

For high-dimensional parameter spaces, as previously discussed, other methods compromise the structural integrity of the original parameter spaces. In contrast, FLoRA preserves their topological structure, leading to superior performance outcomes.

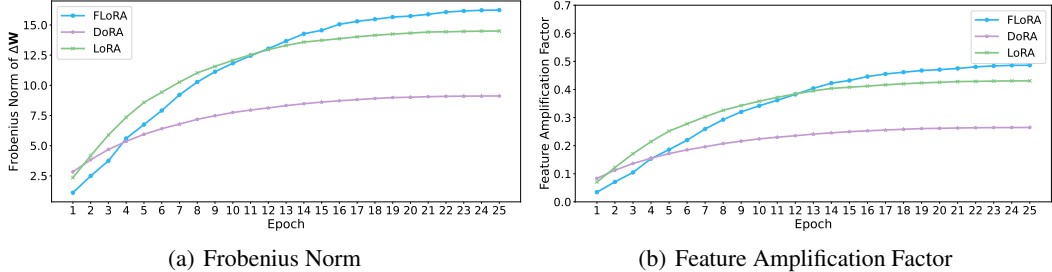

(a) Frobenius Norm          (b) Feature Amplification Factor

Figure 4: Average of the Frobenius norm of $\Delta\mathbf{W}$ and the feature amplification factor during training.

To further substantiate the analysis on FLoRA's superiority, we record the average of the Frobenius norm of $\Delta\mathbf{W}$ and the *feature amplification factor* $\frac{\|\Delta\mathbf{W}\|_F}{\|\mathbf{U}^\mathsf{T}\mathbf{W}\mathbf{V}\|_F}$ Hu et al. (2021a) of all the layers in DeBERTaV3-base on CoLA during training, with FLoRA, DoRA and LoRA's rank $r = 8$. Here $\mathbf{U}$ and $\mathbf{V}$ being the top $r$ left- and right-singular matrices of the SVD decomposition of $\Delta\mathbf{W}$. The feature amplification factor measures how much of task-specific information are amplified by $\Delta\mathbf{W}$.

The results, illustrated in Fig. 4, show that LoRA and DoRA can amplify more task-specific features than FLoRA in the early stages. As discussed before, these two methods have strong constraints on matrix patterns, resulting in a distinct directional learning pattern at the beginning, which contributes to their larger initial values. However, such constraints on matrix patterns may not always be suitable for downstream tasks, and since downstream datasets may possess various properties, their amplification factors upon convergence are smaller than that of FLoRA.

Additionally, we found the trend of the Frobenius norm closely aligns with the feature amplification factor, surprisingly. This might indicate that a larger Frobenius norm of $\Delta\mathbf{W}$ values can accommodate more task-specific information, thereby more effectively amplifying the task-specific information in the frozen weights. We also observed that, initially, the Frobenius norms of $\Delta\mathbf{W}$ for LoRA and DoRA are also larger than that of FLoRA, for they can quickly capture the information due to the learning of specific matrix patterns. However, upon convergence, they are both smaller than FLoRA's, suggesting that FLoRA can contain more kinds of properties for task-specific information.

### 5.3 DOES FLORA REQUIRE ACCEPTABLE TRAINING COSTS COMPARED TO OTHER METHODS?

We assess the training costs under various configurations. All training hyper-parameters, such as batch size and epochs, are kept consistent, with the results presented in Table 5. Clearly, FLoRA is more efficient in terms of training time and memory footprint compared with SOTA method DoRA, which require significantly more time and GPU memories.

Table 5: Training time (minutes/epoch) and GPU usage (GB) of FLoRA.

| Method | ConvNeXt-V2-L | | | | | Method | DeBERTaV3-base | | | | | | | |
|--------|---------------|--------|------|-------|------|--------|----------------|--------|------|------|------|------|------|------|
| | # Params | COCO | | ADE20K | | | # Params | MNLI | | SST-2 | | STS-B | | |
| | | Time | GPU | Time | GPU | | | Time | GPU | Time | GPU | Time | GPU | |
| LoRA | 6.59 % | 70 | 17.19 | 53 | 11.94 | LoRA | | 73.57 | 11.35 | 6.38 | 6.85 | 0.56 | 6.85 | |
| DoRA | 6.59 % | 100 | 30.67 | 67 | 17.36 | DoRA | 0.18 % | 118.42 | 16.72 | 11.26 | 9.66 | 0.91 | 9.66 | |
| FLoRA | 6.50 % | 77 | 17.94 | 60 | 13.10 | FLoRA | | 79.57 | 11.35 | 6.83 | 6.86 | 0.56 | 6.85 | |
| LoRA | 26.36 % | 77 | 18.11 | 52 | 12.79 | LoRA | | 74.29 | 11.39 | 6.45 | 6.88 | 0.59 | 6.87 | |
| DoRA | 26.36 % | 111 | 31.41 | 69 | 17.86 | DoRA | 0.72 % | 117.14 | 16.75 | 11.79 | 9.69 | 0.92 | 9.69 | |
| FLoRA | 26.06 % | 74 | 18.23 | 58 | 13.45 | FLoRA | | 78.57 | 11.39 | 6.67 | 6.88 | 0.60 | 6.87 | |

### 5.4 IS FLORA SENSITIVE TO THE SCALING FACTOR?

We report the sensitivity w.r.t. the scale $s$ on ConvNeXt-V2-L and DeBERTaV3-base, and the results are shown in Fig. 5. It is obvious that the performance of FLoRA is quite stable with the scale changing in a reasonable range, which is a desirable property in practice.

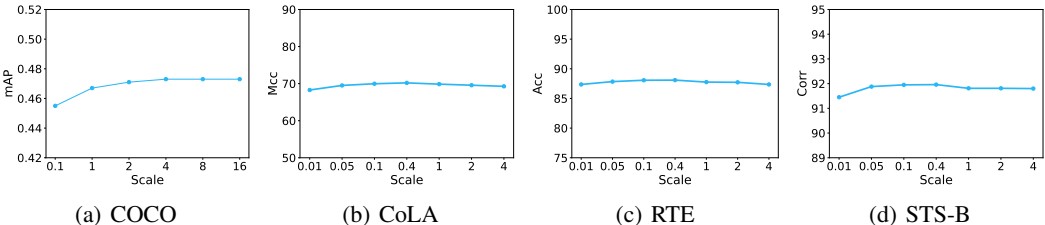

|                (a) COCO                |                (b) CoLA                |                (c) RTE                |                (d) STS-B                |

Figure 5: Sensitivity analysis of FLoRA w.r.t. scale $s$.

## 6 RELATED WORK

### 6.1 PARAMETER-EFFICIENT FINE-TUNING

Methods for Parameter-Efficient Fine-tuning (PEFT) have been conceived to mitigate the substantial computational costs associated with the fine-tuning of large-scale models. This economization is realized by honing a comparatively minute fraction of the overall parameters, selected strategically for adaptation to a variety of downstream tasks. Current PEFT techniques can be divided into three distinct categories Liu et al. (2024b); Ding et al. (2023): Adapter-based Houlsby et al. (2019); Chen et al. (2022); Luo et al. (2023); He et al. (2021a); Mahabadi et al. (2021); Karimi Mahabadi et al. (2021), prompt-based Lester et al. (2021); Razdaibiedina et al. (2023); Wang et al. (2023b); Shi & Lipani (2023); Fischer et al. (2024) and low-rank matrix adaptation-based Hu et al. (2021a); Liu et al. (2024b); Hyeon-Woo et al. (2021); Qiu et al. (2023); Renduchintala et al. (2023); Kopiczko et al. (2023); YEH et al. (2023); Zhang et al. (2022a); Si et al. (2024a). The first category of method integrates linear modules either sequentially or concurrently with the existing layer to enhance the performance, and the second class introduces additional soft tokens (prompts) to the initial input and concentrate exclusively on refining these trainable vectors. The last type, proposed by LoRA Hu et al. (2021a), adopts low-rank matrix adaptation to model the change of the weight during fine-tuning, and are capable of merging with pre-trained weights.

However, these methods only focuses on linear weights or destructing the structural integrity of high dimensional parameter spaces. To this end, we propose a novel method FLoRA to address various dimensional parameter space.

## 6.2 Tucker Decomposition

When designing FLoRA, our goal was to preserve the topological structure of different dimensions of parameter space while maintaining the changes within a low-rank manifold. To achieve this, we use a tensor of the same dimensionality as the original parameter space to model its changes. By learning the corresponding projection matrices, we align the dimensional sizes of the low-rank space with those of the original space. From this perspective, we found that the form of Tucker decomposition Tucker (1966) naturally aligns with our approach.

Tucker decomposition is one of the well-studied algebraic tensor decomposition. Formally, given a tensor $\mathcal{X} \in \mathbb{R}^{I_1 \times I_2 \times \cdots \times I_N}$, where $N$ is the order of the tensor (i.e., the number of dimensions or modes), Tucker decomposition represents $\mathcal{X}$ as a product of a core tensor $\mathcal{G} \in \mathbb{R}^{J_1 \times J_2 \times \cdots \times J_N}$ and a matrix along each mode $n$, $\mathbf{A}^{(n)} \in \mathbb{R}^{I_n \times J_n}$, where $J_n$ can be considered as the dimension of the core tensor along mode $n$. The decomposition can be compactly written as:

$$\mathcal{X} = \mathcal{G} \times_1 \mathbf{A}^{(1)} \times_2 \mathbf{A}^{(2)} \times \cdots \times_N \mathbf{A}^{(N)}, \tag{7}$$

where $\times_n$ denotes the or mode-$n$ product between a tensor and a matrix. The core tensor $\mathcal{G}$ represents the interactions between different modes, and the matrices $\mathbf{A}^{(n)}$ are analogues to principal components within each respective mode. The selection of dimensions $J_1, J_2, \ldots, J_N$ allows for a balance between desired approximation quality and computational efficiency, tailored to the specific requirements of the task at hand.

## 6.3 Other Methods Based on Low-rank Tensor Adaptations

There are some works like LoTR Bershatsky et al. (2024) and SuperLoRA Chen et al. (2024b) also adopt low-rank tensor adaptations, but they are unrelated to FLoRA.

**Usage and Impact.** Based on low-rank tensor adaptation, FLoRA directly models $\Delta\mathbf{W}$ for each weight, maintaining its spatial structure. However, LoTR concatenates $\Delta\mathbf{W}$ across all layers and then applies low-rank tensor adaptation, assuming all layers share same weight matrices, which is a strong assumption that leads to suboptimal performance. SuperLoRA concatenates the vectorized $\Delta\mathbf{W}$ across all Multi-Layer Attention layers, performs various operations, and then redistributes them back to the original dimensions. In this process, low-rank tensor adaptation is just one choice. Besides, both LoTR and SuperLoRA disrupt the original parameter space structure although they use low-rank tensor adaptation.

**Motivation and Details.** FLoRA aims to model $\Delta\mathbf{W}$ while preserving the N-dimensional topological structure, meanwhile addressing the limitations of other methods focusing only on linear weights. However, LoTR aims to achieve extreme parameter efficiency, and SuperLoRA aims to unify and extend LoRA derivatives by setting different hyper-parameters. Both LoTR and SuperLoRA are tailored for linear weights, limiting their applications.

## 7 Conclusion

In this paper, we propose a generalized low-rank tensor adaptation based PEFT method, FLoRA, aiming for N-dimension parameter space. FLoRA asserts that the alternations in each dimensional parameter space contain a low-rank core space structurally consistent to the original space. It models updates using this core space alongside corresponding weights to reconstruct the original alternation space. By doing so, FLoRA effectively preserves the structural integrity of the original N-dimensional parameter space while modeling its change through low-rank tensor adaptations. Extensive experiments on computer vision, natural language processing and multi-modal domains substantiate the effectiveness of FLoRA.

There still exists some limitations for FLoRA. For a specific backbone, FLoRA could achieve stable superior performance on different datasets when the scaling factor $s$ changing within a wide range. While for different backbones such as ConvNeXt-V2-L, InternViT-6B and DeBERTaV3-base, it still needs different scales. Understanding the role of scale in different models and designing a unified scale is a topic worthy of further investigation.

ACKNOWLEDGEMENTS

This work was supported by NSFC 62322604, NSFC 62176159, NSFC 62401361, and Shanghai Municipal Science and Technology Major Project 2021SHZDZX0102.

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

# A MORE EXPERIMENTS

## A.1 GENERATIVE TASKS

We employ FLoRA and LoRA to fine-tune Stable Diffusion Rombach et al. (2022) SDXL with the toolkit diffusers supported by HuggingFace using the AdamW optimizer Loshchilov & Hutter (2017), with a learning rate of $1 \times 10^{-4}$. We fine-tune all the convolutional layers of the U-net Ronneberger et al. (2015). Specifically, we fine-tune SDXL to learn the style of Pokemon trained on Pokemon BLIP captions dataset. We show some visualization results of FLoRA and LoRA conditioned on the same texts prompts in Fig. 6. The results indicate that FLoRA outperforms LoRA in generating images that adhere more closely to the desired style. This can be attributed to the more sophisticated parameter fine-tuning of FLoRA.

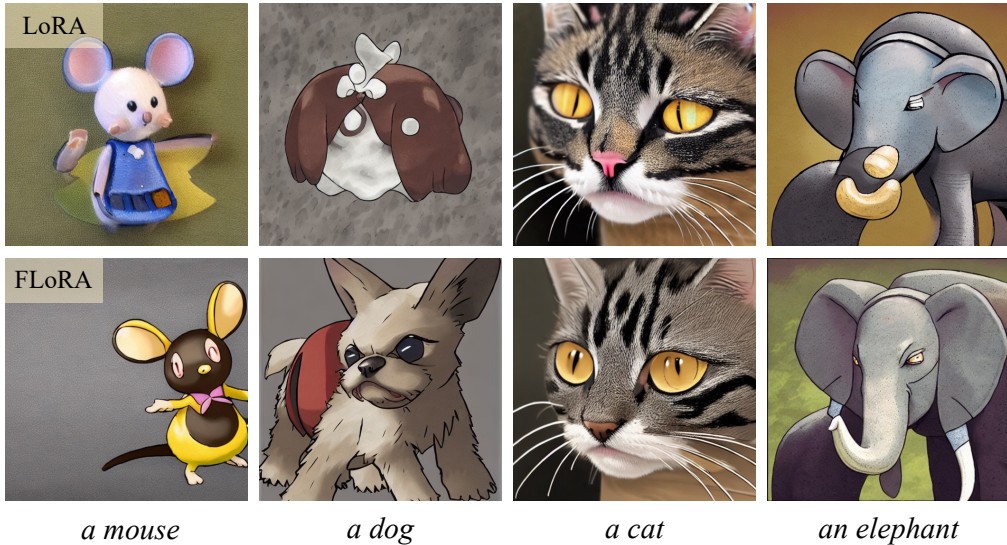

| a mouse | a dog | a cat | an elephant |

Figure 6: The generative results fine-tuned by LoRA and FLoRA on the Pokemon BLIP captions dataset. FLoRA follows the Pokemon style more closely.

## A.2 COMMONSENSE REASONING TASKS

The commonsense reasoning benchmarks include eight distinct sub-tasks, each paired with a specific dataset: BoolQ Clark et al. (2019), PIQA Bisk et al. (2020), SIQA Sap et al. (2019), HellaSwag Zellers et al. (2019), WinoGrande Sakaguchi et al. (2021), ARC-e/ARC-c Clark et al. (2018), and OBQA Mihaylov et al. (2018). Following the approach in Hu et al. (2023), we combine the training datasets from these tasks into a single Commonsense170K dataset and evaluate performance on the test set of each task.

We use FLoRA to fine-tune LLaMA3-8B AI@Meta (2024) on commonsense reasoning task. Additionally, we incorporate results from ChatGPT's gpt-3.5-turbo API using zero-shot Chain of Thought reasoning Wei et al. (2022). The hyper-parameter settings for FLoRA are detailed in Table 6.

The results are shown in Table 7. We can observe that FLoRA outperforms other state-of-the-art methods on this task, demonstrating its exceptional performance.

## A.3 VTAB-1K BENCHMARK

We also evaluate the performance of FLoRA on VTAB-1K Benchmark. VTAB-1K Zhai et al. (2019) includes 19 image classification tasks spanning diverse domains, categorized into three groups: Natural, Specialized, and Structured. These tasks represent a wide variety of potential downstream applications, making this benchmark a strong indicator of a method's transfer learning capability. Each dataset consists of 800 training samples and 200 validation samples. Following prior studies Jia et al. (2022); Jie & Deng (2022; 2023), we fine-tune the pre-trained model ViT-B/16 Dosovitskiy

Table 6: Hyper-parameter configurations for commonsense reasoning task.

| Hyper-parameter | LoRA | AdaLoRA | DoRA | FLoRA |
|---|---|---|---|---|
| Rank r | | 16 | | |
| $\alpha$ | | 32 | | |
| Dropout | | 0.05 | | |
| Batch size | | 16 | | |
| Epochs | | 3 | | |
| Learning rate | | 3e-5 | | |
| Target module | | *q, k, v, up, down* | | |

Table 7: Performance comparison across commonsense reasoning tasks. Average (Avg.) performance is computed over all tasks.

| Method | # Params (%) | BoolQ | PIQA | SIQA | HellaSwag | WinoGrande | ARC-e | ARC-c | OBQA | Avg. |
|---|---|---|---|---|---|---|---|---|---|---|
| ChatGPT | - | 73.1 | 85.4 | 68.5 | 78.5 | 66.1 | 89.8 | 79.9 | 74.8 | 77.0 |
| LoRA | 0.35 | 72.3 | 86.7 | 79.3 | 93.5 | 84.8 | 87.7 | 75.7 | 82.8 | 82.9 |
| AdaLoRA | 0.35 | 75.1 | 86.4 | 76.7 | 75.4 | 83.3 | 90.4 | 79.1 | 85.0 | 81.4 |
| DoRA | 0.36 | 74.5 | 88.8 | 80.3 | 95.5 | 84.7 | 90.1 | 79.1 | 87.2 | 85.0 |
| FLoRA | 0.35 | 74.6 | 89.7 | 81.0 | 95.0 | 85.8 | 90.5 | 81.5 | 86.8 | 85.6 |

et al. (2021) using all 1,000 training and validation samples and evaluate performance on the test set. Consistent with Jia et al. (2022); Lian et al. (2022), we use unnormalized inputs, aligning with the original VTAB paper Zhai et al. (2019).

We compare FLoRA with SOTA methods and some CV tailored methods, including VPT Jia et al. (2022), NOAH Zhang et al. (2022b), SSF Lian et al. (2022), PAdapter Pfeiffer et al. (2020), FACT Jie & Deng (2023), Adaptformer Chen et al. (2022), Compacter Karimi Mahabadi et al. (2021), LoRA Hu et al. (2021a) and BitFit Zaken et al. (2021). All baselines are evaluated using FP32 precision by default. For PAdapter, Adaptformer, and LoRA, the hidden dimension is set to 8, and FLoRA 2. For Compacter, the number of Kronecker products and the hidden dimensions are set to 4 and 32, respectively. FACT employs the FACT-TT variant with the rank selected from 8, 16, 32 to adapt the MHSA blocks. The configurations for other baselines follow their respective original papers.

Clearly, compared to methods specifically designed for computer vision tasks, FLoRA continues to demonstrate significant advantages.

Table 8: Performance comparison across VTAB-1K benchmarks. Results are grouped into Natural, Specialized, and Structured categories, with average accuracy (%) listed. Avg. denotes the average results over three groups. #P: number of parameters D1: Cifar100. D2: Caltech101. D3: DTD. D4: Flower102. D5: Pets. D6: SVHN. D7: Sun397. D8: Camelyon. D9: EuroSAT. D10: Resisc45. D11: Retinopathy. D12: Clevr-Count. D13: Clevr-Dist. D14: DMLab. D15:KITTI-Dist. D16: dSpr-Loc. D17: dSpr-Ori. D18: sNORB-Azim. D19: sNORB-Ele.

| Method | #P (M) | Avg | Natural | | | | | | | Specialized | | | | Structured | | | | | | | |
|---|---|---|---|---|---|---|---|---|---|---|---|---|---|---|---|---|---|---|---|---|---|
| | | | D1 | D2 | D3 | D4 | D5 | D6 | D7 | D8 | D9 | D10 | D11 | D12 | D13 | D14 | D15 | D16 | D17 | D18 | D19 |
| Full | 85.8 | 68.9 | 68.9 | 87.7 | 64.3 | 97.2 | 86.9 | 87.4 | 38.8 | 79.7 | 95.7 | 84.2 | 73.9 | 56.3 | 58.6 | 41.7 | 65.5 | 57.5 | 46.7 | 25.7 | 29.1 |
| Last-layer Tuning | 0.08 | 57.6 | 64.4 | 85.0 | 63.2 | 97.0 | 86.3 | 36.6 | 51.0 | 78.5 | 87.5 | 68.5 | 74.0 | 34.3 | 30.6 | 33.2 | 55.4 | 12.5 | 20.0 | 9.6 | 19.2 |
| VPT-Deep | 2.03 | 72.0 | 78.8 | 90.8 | 65.8 | 98.0 | 88.3 | 78.1 | 49.6 | 81.8 | 96.1 | 83.4 | 68.4 | 68.5 | 60.0 | 46.5 | 72.8 | 73.6 | 47.9 | 32.9 | 37.8 |
| NOAH | 1.37 | 75.5 | 69.6 | 92.7 | 70.2 | 99.1 | 90.4 | 86.1 | 53.7 | 84.4 | 95.4 | 83.9 | 75.8 | 82.8 | 68.9 | 49.9 | 81.7 | 81.8 | 48.3 | 32.8 | 44.2 |
| LoRA | 1.13 | 76.4 | 72.0 | 91.2 | 71.6 | 99.1 | 91.3 | 88.9 | 56.4 | 87.2 | 94.6 | 83.9 | 74.9 | 83.7 | 64.0 | 52.3 | 81.2 | 84.8 | 53.3 | 38.1 | 43.4 |
| SSF | 0.78 | 75.7 | 69.0 | 92.6 | 75.1 | 99.4 | 91.8 | 90.2 | 52.9 | 87.4 | 95.9 | 87.4 | 75.5 | 75.9 | 62.3 | 53.3 | 80.6 | 77.3 | 54.9 | 29.5 | 37.9 |
| PAdapter | 0.56 | 75.5 | 73.2 | 90.1 | 69.6 | 99.2 | 91.1 | 84.9 | 56.0 | 86.6 | 94.8 | 82.5 | 75.8 | 82.9 | 63.9 | 49.7 | 79.7 | 81.7 | 55.5 | 31.6 | 42.2 |
| AdaptFormer | 0.56 | 76.7 | 73.8 | 92.3 | 72.7 | 99.3 | 91.6 | 89.1 | 56.5 | 87.8 | 95.5 | 84.9 | 75.2 | 83.3 | 62.5 | 52.4 | 81.7 | 86.2 | 55.9 | 34.4 | 40.2 |
| BitFit | 0.39 | 65.2 | 72.8 | 87.0 | 59.2 | 97.5 | 85.3 | 59.9 | 51.4 | 78.7 | 91.6 | 72.9 | 69.8 | 61.5 | 55.6 | 32.4 | 55.9 | 66.6 | 40.0 | 15.7 | 25.1 |
| FACT-TT | 0.30 | 76.7 | 73.4 | 91.0 | 72.4 | 99.2 | 91.4 | 90.1 | 56.6 | 87.3 | 94.7 | 84.5 | 75.8 | 83.0 | 64.9 | 51.3 | 81.4 | 87.4 | 53.2 | 33.5 | 44.3 |
| VPT-Shallow | 0.24 | 67.8 | 77.7 | 86.9 | 62.6 | 97.5 | 87.3 | 74.5 | 51.2 | 78.2 | 92.0 | 75.6 | 72.9 | 50.5 | 58.6 | 40.2 | 67.1 | 68.7 | 36.1 | 20.2 | 34.1 |
| Compacter | 0.15 | 74.2 | 71.9 | 89.0 | 69.7 | 99.1 | 90.7 | 82.7 | 56.1 | 86.0 | 93.5 | 82.4 | 75.3 | 80.2 | 63.4 | 47.4 | 77.2 | 78.1 | 53.5 | 27.3 | 39.8 |
| FLoRA | 0.32 | **77.3** | 73.8 | 92.1 | 73.2 | 99.3 | 91.4 | 89.9 | 56.4 | 88.1 | 95.8 | 86.8 | 76.2 | 83.9 | 65.5 | 52.9 | 81.4 | 86.2 | 54.8 | 35.2 | 42.4 |

## B   DETAILED DISCUSSION ON SEC. 5.2

### B.1   THEORETICAL INSIGHTS ON THE EFFECTIVENESS OF FLORA'S REPRESENTATION

Inspired by Si et al. (2024b), we provide the theoretical insights on the effectiveness of FLoRA's low-rank representation here.

Let $\mathbf{W}^* \in \mathbb{R}^{n \times m}$ be the optimal weight for a specific linear layer with frozen weight $\mathbf{W}$, and $\Delta\mathbf{W}^* = \mathbf{W}^* - \mathbf{W}$ is the optimal change. Generally, the low-rank representation of LoRA's series can be represented as $\Delta\mathbf{W} = \mathbf{AGB}$, where $\mathbf{A} \in \mathbb{R}^{n \times r}$, $\mathbf{B} \in \mathbb{R}^{r \times m}$, $\mathbf{G} \in \mathbb{R}^{r \times r}$ and $r \ll \{n, m\}$, and the essential difference of these methods is the matrix pattern of $\mathbf{G}$. In FLoRA, $\mathbf{G}$ is arbitrary. For other methods such as AdaLoRA, $\mathbf{G}$ is diagonal, and identity for LoRA.

Ideally, we have $\mathbf{G} = \mathbf{A}^\dagger \Delta\mathbf{W}^* \mathbf{B}^\dagger$, where $\mathbf{A}^\dagger$ and $\mathbf{B}^\dagger$ are the Moore-Penrose Pseudo-inverses of $\mathbf{A}$ and $\mathbf{B}$, respectively. If $\mathbf{G}$ is an arbitrary matrix such as FLoRA, it is obvious that no constraints are imposed on $\mathbf{A}$ and $\mathbf{B}$, since for any choice of $\mathbf{A}$, $\mathbf{B}$, we can always find a $\mathbf{G}$. However, when $\mathbf{G}$ is diagonal or identity, $\mathbf{A}$ and $\mathbf{B}$ are inevitably subject to certain constraints, and there may even exist dependencies or interrelations between them.

Thus, to achieve the optimal change $\Delta\mathbf{W}^*$, FLoRA imposes no constraints on $\mathbf{A}$, $\mathbf{B}$. Other methods, to varying degrees, impose certain constraints on $\mathbf{A}$ and $\mathbf{B}$. However, these constraints are not applied during training, which means that these methods like LoRA are less likely to learn the optimal change compared to FLoRA. Further, even if specific matrix patterns are constrained during training, FLoRA's greater degree of freedom and stronger expressive power in learning make it much easier to achieve better results Zhang et al. (2022a); Hu et al. (2021b). Additionally, some subsequent works have also demonstrated the effectiveness of FLoRA from different perspectives Wu et al. (2024).

### B.2   MORE EXPLANATION ON THE EMPIRICAL PHENOMENA

#### B.2.1   WHAT IS FEATURE AMPLIFICATION FACTOR?

The Feature Amplification Factor is a metric introduced by LoRA to measure how much task-specific information is amplified by $\Delta\mathbf{W}$. Simply put, it calculates the ratio of the task-specific information learned by the model (represented by the Frobenius norm of $\Delta\mathbf{W}$) to the amount of information obtained by projecting the pre-trained weights $\mathbf{W}$ onto the task-specific directions. Mathematically, the Feature Amplification Factor is defined as $\frac{\|\Delta\mathbf{W}\|_F}{\|\mathbf{U}^\top \mathbf{W} \mathbf{V}\|_F}$, where $\mathbf{U}$ and $\mathbf{V}$ are the top $r$ left and right singular vectors of $\Delta\mathbf{W}$. Here, $r$ is the rank configuration of PEFT methods like LoRA.

This metric provides insight into how effectively the model adapts to downstream tasks by quantifying the relative contribution of the learned task-specific updates, and it has been widely adopted in PEFT works to evaluate the amount of task-specific information learned by the model Wang et al. (2024); Si et al. (2024a); Wang & Li (2024).

#### B.2.2   RELATIONSHIP BETWEEN FEATURE AMPLIFICATION FACTOR AND PERFORMANCE

Indeed, although LoRA does not explicitly state a relationship between these factors and task performance, its experimental results show a clear correlation with amplification levels. Moreover, many existing works Wang et al. (2024); Si et al. (2024a); Wang & Li (2024) implicitly or explicitly suggest that a larger Feature Amplification Factor correlates with improved model performance. For instance, Si et al. (2024a) directly highlights that amplifying more task-specific information can enhance model performance.

#### B.2.3   RELATIONSHIP BETWEEN FEATURE AMPLIFICATION FACTOR AND FROBENIUS NORM OF $\Delta\mathbf{W}$

In our experiments, we observed that the final value of the Frobenius norm of $\Delta\mathbf{W}$ is positively correlated with the Feature Amplification Factor, meaning that as one increases, the other also tends to increase. This was somewhat unexpected but also intuitively reasonable: given that the Feature Amplification Factor is directly calculated using the Frobenius norm of $\Delta\mathbf{W}$, it is logical to assume some intrinsic connection between the two. Here we offer an explanation for this phenomenon:

Firstly, the Feature Amplification Factor is defined as the ratio of the Frobenius norm of $\Delta\mathbf{W}$, i.e., $\|\Delta\mathbf{W}\|_F$, to the amount of information in the pre-trained weights $\mathbf{W}$ projected onto the task-specific directions, denoted as $\|\mathbf{U}^\mathsf{T}\mathbf{W}\mathbf{V}\|_F$. Since the pre-trained weight matrix $\mathbf{W}$ is frozen during training, the value of $\|\mathbf{U}^\mathsf{T}\mathbf{W}\mathbf{V}\|_F$ depends solely on $\mathbf{U}$ and $\mathbf{V}$, which are two orthonormal matrices derived from $\Delta\mathbf{W}$ and represent task-specific directions.

In Si et al. (2024a), the authors highlight that at any step during training, $\Delta\mathbf{W}$ effectively captures the task-specific directions. Consequently, from the beginning to the end of training, the changes in $\mathbf{U}$ and $\mathbf{V}$ are minimal (as also validated in Si et al. (2024a)), which implies that $\|\mathbf{U}^\mathsf{T}\mathbf{W}\mathbf{V}\|_F$ remains nearly constant. In other words, a larger $\|\Delta\mathbf{W}\|_F$ does indeed correspond to a larger Feature Amplification Factor.

Building on the analysis of the relationship between amplification factors and model performance, it is evident that a larger Frobenius norm of $\Delta\mathbf{W}$ corresponds to better performance. This is a surprising yet logical and exciting conclusion. We hypothesized that larger Frobenius norm values of $\Delta\mathbf{W}$ allow for encoding more task-specific information, thereby effectively amplifying the task-specific signals in the frozen weights. Upon further reflection, this insight aligns with the findings of Si et al. (2024a) but from a different perspective: amplifying task-specific information leads to better task performance. Moreover, directly increasing the Frobenius norm of $\Delta\mathbf{W}$ emerges as a more straightforward approach to achieve this amplification.

Therefore, we believe that exploring more effective PEFT methods from this perspective represents a highly promising direction for future research.

### B.2.4 MORE EXAMPLES

To further support the above analysis, we record the average of the Frobenius norm of $\Delta\mathbf{W}$ and the feature amplification factor of all the layers in DeBERTaV3-large on SST2 during training, with FLoRA, DoRA and LoRA's rank $r = 8$, and the results are illustrated in Fig. 7. Clearly, the results align well with our aforementioned analysis, further reinforcing our conclusions.

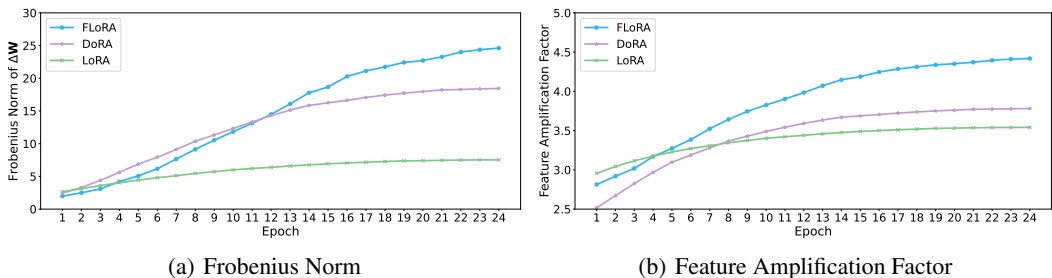

(a) Frobenius Norm          (b) Feature Amplification Factor

Figure 7: Average of the Frobenius norm of $\Delta\mathbf{W}$ and the feature amplification factor during training.

## C DETAILS ON TRAINING

### C.1 TRAINING DETAILS FOR CV TASKS

Table 9 presents some customized hyper-parameters for all experiments of FLoRA, LoRA Hu et al. (2021a), DoRA Liu et al. (2024b), BitFit Zaken et al. (2021) and FT. Please refer to MMDetection / MMSegmentation / MMRotate for other default parameters we omit in Table 9.

Table 9: Hyper-parameter setup for object detection and segmentation.

| Dataset | Toolkit | Model | Schedule | LR | BS | Optimizer | Weight Decay |
|---|---|---|---|---|---|---|---|
| COCO | MMDetection Chen et al. (2019) | Mask R-CNN He et al. (2017) | 12ep | 1e-4 | 32 | AdamW | 5e-2 |
| DOTA | MMRotate Zhou et al. (2022) | Oriented R-CNN Xie et al. (2021) | 12ep | 5e-3 | 16 | SGD | 1e-4 |
| ADE20K | MMSegmentation Contributors (2020) | UperNet Xiao et al. (2018) | 160k | 1e-4 | 16 | AdamW | 5e-2 |

## C.2 DETAILS OF GLUE DATASET

We present the details of the GLUE dataset, which are shown in Table 10.

Table 10: Details of GLUE dataset.

| Dataset | Task | # Train | # Dev | # Test | # Label | Metrics |
|---------|------|---------|-------|--------|---------|---------|
| Single-Sentence Classification | | | | | | |
| CoLA | Acceptability | 8.5k | 1k | 1k | 2 | Matthews corr |
| SST | Sentiment | 67k | 872 | 1.8k | 2 | Accuracy |
| Pairwise Text Classification | | | | | | |
| MNLI | NLI | 393k | 20k | 20k | 3 | Accuracy |
| RTE | NLI | 2.5k | 276 | 3k | 2 | Accuracy |
| QQP | Paraphrase | 364k | 40k | 391k | 2 | Accuracy / F1 |
| MRPC | Paraphrase | 3.7k | 408 | 1.7k | 2 | Accuracy / F1 |
| QNLI | QA/ NLI | 108k | 5.7k | 5.7k | 2 | Accuracy |
| Text Similarity | | | | | | |
| STS-B | Similarity | 7k | 1.5k | 1.4k | 1 | Pearson/ Spearman Corr |

## C.3 TRAINING DETAILS FOR NLP TASKS

We provide the training details for NLP tasks in Table 11.

Table 11: Hyper-parameter setup for GLUE benchmark.

| Dataset | MNLI | RTE | QNLI | MRPC | QQP | SST-2 | CoLA | STS-B |
|---------|------|-----|------|------|-----|-------|------|-------|
| learning rate | $5 \times 10^{-4}$ | $1.2 \times 10^{-3}$ | $1.2 \times 10^{-3}$ | $1 \times 10^{-3}$ | $5 \times 10^{-4}$ | $8 \times 10^{-4}$ | $5 \times 10^{-4}$ | $2.2 \times 10^{-3}$ |
| batch size | 32 | 32 | 32 | 32 | 32 | 32 | 32 | 32 |
| #epochs | 7 | 50 | 5 | 30 | 5 | 24 | 25 | 25 |

## C.4 TRAINING DETAILS FOR MULTI-MODAL TASKS

We provide the training details for multi-modal tasks here in Table 12.

Table 12: Hyper-parameter configurations on FLoRA for fine-tuning LLaVA-1.5-7B with visual instruction tuning datasets.

| Hyper-parameters | Value |
|------------------|-------|
| Rank $r$ | 128 |
| $s$ | 2 |
| Dropout | 0.05 |
| Optimizer | AdamW |
| LR | $2 \times 10^{-4}$ |
| LR Scheduler | Cosine decay |
| Batch size | 16 |
| Warmup ratio | 0.03 |
| Epochs | 1 |
| Where | Q,K,V,O,Up,Down,Gate |

