# OpenReview forum: "Maintaining Structural Integrity in Parameter Spaces for Parameter Efficient Fine-tuning"
_ICLR.cc/2025/Conference — ICLR 2025 Poster_

### Official Review · Reviewer_wiu6 · 2024-10-29

**Soundness:** 2
**Presentation:** 2
**Contribution:** 2
**Rating:** 5
**Confidence:** 3

**Summary:**

This paper presents FLoRA, a novel parameter-efficient fine-tuning framework that addresses the challenge of adapting pre-trained models across diverse parameter space dimensions. Leveraging Tucker decomposition, FLoRA proposes to model parametric changes through a low-rank core space, ostensibly preserving the structural integrity of high-dimensional parameter spaces. The methodology is evaluated across a spectrum of tasks encompassing computer vision, natural language processing, and multimodal learning. The authors report that FLoRA demonstrates superior performance compared to existing methods, notably LoRA, while utilizing fewer trainable parameters. This efficiency is attributed to FLoRA's capacity to maintain the topological structure of the parameter space.

**Strengths:**

1)	The motivation behind this paper is significant. When using LoRA to fine-tune convolutional layers, it either disrupts the original topology or adds too many parameters. FLoRA, with Tucker decomposition, effectively adjusts high-dimensional parameter spaces, aiding the application of parameter-efficient fine-tuning across more models.
2)	Although FLoRA is motivated by improvements in tuning high-dimensional parameter spaces, the paper still includes many experiments with models that mainly use linear layers. The experimental setup of the paper covering many fields including computer vision, NLP and multimodal tasks, and shows good performance.
3)	The authors compare FLoRA with LoRA and DoRA, using both theoretical reasoning and empirical evidence. They employ metrics like Frobenius norm and feature amplification factor, offering insights into learning patterns. Notably, they explore the correlation between Frobenius norm and task-specific information amplification.

**Weaknesses:**

1)	The performance gains of FLoRA on linear layer-based models appear limited. While the motivation to address topological distortions in high-dimensional parameter spaces is commendable, it potentially constrains FLoRA's applicability across diverse model architectures. Although the analysis utilizing Frobenius norm metrics and unrestricted low-rank subspaces is insightful, I recommend enhancing the paper through: A more rigorous theoretical treatment, exploring deeper mathematical foundations. An expanded experimental paradigm encompassing various model architectures, tasks, and datasets. These enhancements would strengthen the generalizability claims and broaden FLoRA's potential impact in the field of parameter-efficient fine-tuning.
2)	The initialization strategy in FLoRA deserves more exploration. Although the authors use a conservative approach, diverse methods could enhance performance. Advances in LoRA initialization indicate room for improvement. A study comparing various strategies might boost FLoRA's effectiveness and stability across tasks and models. I look forward to the authors expanding research on how initialization affects FLoRA's low-rank subspace, potentially leading to more robust fine-tuning.

**Questions:**

See above

---

> ### Author Response · Authors · 2024-11-17
> **Response to Reviewer wiu6**
>
> Dear Reviewer wiu6:
>
> We would like to first extend our sincere gratitude for your time and effort in evaluating our manuscript. Your thorough evaluation and insightful comments are greatly appreciated. We will address your questions point by point and hope to resolve your concerns effectively.
>
>
>
> ## **Weakness 1, More Theoretical Proofs and Experiments**
>
> We sincerely appreciate your suggestions on how to improve our manuscript regarding linear layer. We hope the following responses address your concerns effectively.
>
> ### Theoretical Proof
>
> Let $\\mathbf{W}^*\\in\\mathbb{R}^{n\times m}$ be the optimal weight for a specific linear layer with frozen weight $\\mathbf{W}$, and $\\Delta\\mathbf{W}^{\*}=\\mathbf{W}^{\*}-\\mathbf{W}$ is the optimal change. Generally, the low-rank representation of LoRA’s series can be represented as $\\Delta\\mathbf{W}=\\mathbf{AGB}$, where $\\mathbf{A}\\in\\mathbb{R}^{n\\times r}$, $\\mathbf{B}\\in\\mathbb{R}^{r\\times m}$, $\\mathbf{G}\\in\\mathbb{R}^{r\times r}$ and $r\\ll\{n,m\}$, and the essential difference of these methods is the matrix pattern of $\\mathbf{G}$. In FLoRA, $\\mathbf{G}$ is arbitrary, in AdaLoRA, it is diagonal, and identity in LoRA.
>
> Ideally, we have $\\mathbf{G}=\\mathbf{A}^{\\dagger}\\Delta\\mathbf{W}^{\*}\mathbf{B}^{\\dagger}$, where $\\mathbf{A}^{\\dagger}$ and $\\mathbf{B}^{\\dagger}$ are the Moore-Penrose Pseudo-inverses of $\\mathbf{A}$ and $\\mathbf{B}$, respectively. If $\\mathbf{G}$ is an arbitrary matrix such as FLoRA, it is obvious that no constraints are imposed on $\\mathbf{A}$ and $\\mathbf{B}$, since for any choice of $\\mathbf{A}$, $\\mathbf{B}$, we can always find a $\\mathbf{G}$. Considering when $\\mathbf{G}$ is diagonal or identity, we use SVD on $\\Delta\\mathbf{W}^{\*}=\\mathbf{U} \\mathbf{\\Sigma} \\mathbf{V}^\\mathsf{T}$ and $\\mathbf{G}=\\mathbf{A}^{\\dagger}\\mathbf{U}\\mathbf{\Sigma}\\mathbf{V}^\\mathsf{T}\\mathbf{B}^{\\dagger}$. We then require $\\mathbf{A}^{\\dagger}=\\mathbf{D}\_1\\mathbf{U}^\\mathsf{T}$ and $\\mathbf{B}^{\\dagger}=\\mathbf{V}\\mathbf{D}\_2$, where $\\mathbf{D}\_1$ and $\mathbf{D}_2$ are rectangular diagonal matrices. It means $\\mathbf{G}=\\mathbf{D}_1\\mathbf{\\Sigma}\\mathbf{D}_2$. If $\\mathbf{G}$ is diagonal like AdaLoRA, we can constrain $\\mathbf{D}\_1=\\mathbf{I}\_{r\\times n}$ and leave $\\mathbf{D}_2$ unrestricted, since any $\\mathbf{D}_2$ can correspond to a $\\mathbf{G}$. However, if $\\mathbf{G}$ is identity such as LoRA, and with $\\mathbf{D}_1=\\mathbf{I}\_{r\\times n}$,  $\\mathbf{D}\_2$ must be $(\\mathbf{D}\_1\\mathbf{\\Sigma})^\\dagger$.
>
> Thus, to achieve the optimal change $\\Delta\\mathbf{W}^\*$, FLoRA imposes no constraints on $\\mathbf{A}$, $\\mathbf{B}$, AdaLoRA has constraints on $\\mathbf{A}$, and LoRA further restricts $\\mathbf{A}$ and $\\mathbf{B}$. However, these constraints are not applied during training, which means that LoRA and AdaLoRA are less likely to learn the optimal change compared to FLoRA. Further, even if specific matrix patterns are constrained during training, FLoRA’s greater degree of freedom and stronger expressive power in learning make it much easier to achieve better results [1,2].
>
> Additionally, some subsequent works have also demonstrated the effectiveness of FLoRA from different perspectives [3,4].
>
> [1] 2023 ICLR, AdaLoRA: Adaptive Budget Allocation for Parameter-Efficient Fine-Tuning
>
> [2] 2024 KNOWL INF SYST, Model Complexity of Deep Learning: A Survey
>
> [3] 2024, arxiv, See Further for Parameter Efficient Fine-tuning by Standing on the Shoulders of Decomposition
>
> [4] 2024, EMNLP, Mixture-of-Subspaces in Low-Rank Adaptation
>
> ### **More Experiments**
>
> We here report the performance of FLoRA when fine-tuning LLAMA-3-8B on commonsense reasoning tasks. As seen in the following table, FLoRA still significantly outperforms LoRA when fine-tuning LLAMA-3-8B.
>
> | Method  | # Params (%) | BoolQ | PIQA | SIQA | HellaSwag | WinoGrande | ARC-e | ARC-c | OBQA | Avg. |
> | ------- | ------------ | ----- | ---- | ---- | --------- | ---------- | ----- | ----- | ---- | ---- |
> | ChatGPT | -            | 73.1  | 85.4 | 68.5 | 78.5      | 66.1       | 89.8  | 79.9  | 74.8 | 77.0 |
> | LoRA    | 0.35         | 72.3  | 86.7 | 79.3 | 93.5      | 84.8       | 87.7  | 75.7  | 82.8 | 82.9 |
> | AdaLoRA | 0.35         | 75.1  | 86.4 | 76.7 | 75.4      | 83.3       | 90.4  | 79.1  | 85.0 | 81.4 |
> | DoRA    | 0.36         | 74.5  | 88.8 | 80.3 | 95.5      | 84.7       | 90.1  | 79.1  | 87.2 | 85.0 |
> | FLoRA   | 0.35         | 74.6  | 89.7 | 81.0 | 95.0      | 85.8       | 90.5  | 81.5  | 86.8 | 85.6 |

---

> ### Author Response · Authors · 2024-11-17
> **Response to Reviewer wiu6**
>
> ## **Weakness 2, Initialization Strategy**
>
> For linear layers, FLoRA requires the initialization of matrices $\mathbf{A}$, $\mathbf{G}$, and $\mathbf{B}$, with one of them being initialized to zero. We focus on the core matrix $\mathbf{G}$. If $\mathbf{G}$ is initialized to zero while $\mathbf{A}$ and $\mathbf{B}$ are randomly initialized, the results are shown in Line 1 of the table. If $\mathbf{G}$ is not initialized to zero, we consider alternative initializations such as an identity matrix, normal distribution, or orthogonal matrix, with the corresponding experimental results shown in Lines 2, 3, and 4, respectively. It can be observed that different initializations of $\mathbf{G}$ have some impact on performance. However, the influence remains within an acceptable range.
>
> | Line | Initialization | BoolQ | PIQA | SIQA | HellaSwag | WinoGrande | ARC-e | ARC-c | OBQA | Avg. |
> | ---- | -------------- | ----- | ---- | ---- | --------- | ---------- | ----- | ----- | ---- | ---- |
> | 1    | zero           | 74.6  | 89.7 | 81.0 | 95.0      | 85.8       | 90.5  | 81.5  | 86.8 | 85.6 |
> | 2    | identity       | 74.4  | 88.9 | 80.7 | 95.1      | 85.3       | 90.0  | 80.5  | 85.4 | 85.0 |
> | 3    | normal         | 75.1  | 88.4 | 80.9 | 95.3      | 85.2       | 90.4  | 80.9  | 85.4 | 85.2 |
> | 4    | orthogonal     | 74.5  | 88.8 | 80.3 | 95.5      | 84.7       | 90.1  | 79.1  | 87.2 | 85.0 |
>
>
> We sincerely hope that our response could address your concerns.

---

> > ### Author Response · Authors · 2024-12-01
> >
> > Dear Reviewer wiu6,
> >
> > May we kindly ask if our responses have addressed your concerns? We look forward to further discussions and feedback from you!
> >
> > Best Regards,
> >
> > Authors.

---

### Official Review · Reviewer_VXNC · 2024-10-30

**Soundness:** 3
**Presentation:** 3
**Contribution:** 3
**Rating:** 6
**Confidence:** 4

**Summary:**

The paper introduces FLoRA as a new approach to Parameter-Efficient Fine-Tuning (PEFT), focusing on preserving the structural integrity of high-dimensional spaces in large models.  The authors argue that existing PEFT methods, especially those relying on low-rank matrix adaptations, often disrupt the crucial spatial relationships within these high-dimensional spaces, particularly in convolutional layers, leading to suboptimal performance.

FLoRA addresses this issue by introducing a low-rank core space that maintains the original spatial dimensions of the parameter space being adapted. For example, for a convolutional layer, FLoRA employs a 4D core tensor that mirrors the structure of the convolutional kernel. This ensures that spatial locality, crucial for convolution operations, is preserved during fine-tuning.

The core space is then combined with corresponding weights to reconstruct the changes in the original parameter space. This approach allows FLoRA to achieve high performance while being efficient in terms of trainable parameters. The paper demonstrates FLoRA's effectiveness across various tasks in computer vision, natural language processing, and multi-modal learning, showcasing its superior performance compared to existing PEFT methods.

* **FLoRA outperforms traditional low-rank methods like LoRA, especially in tasks involving convolutional layers.** This is attributed to FLoRA's ability to preserve the spatial structure of high-dimensional parameter spaces.
* **FLoRA achieves performance comparable to or even exceeding that of full fine-tuning, while using significantly fewer trainable parameters.** This makes it a highly efficient and effective method for adapting large models to downstream tasks.
* **The paper provides empirical evidence for the existence of a low-rank core space within different dimensional parameter spaces.** This supports FLoRA's central premise and suggests its potential for broader applicability.

**Strengths:**

* **Significant Performance Gains:** The paper convincingly demonstrates FLoRA's superior performance over existing PEFT methods on both ConvNeXt (Table 1) and InternViT (Table 2) architectures.  The improvements are particularly noteworthy on ConvNeXt, which relies heavily on convolutional layers, highlighting FLoRA's strength in handling high-dimensional parameter spaces.  The results show that FLoRA not only outperforms other low-rank adaptation methods like LoRA and DoRA by a substantial margin (at least 15% on average across different parameter budgets) but also achieves performance comparable to, and in some cases even better than, full fine-tuning.

* **Parameter Efficiency and Practicality:** The authors emphasize FLoRA's practicality by showcasing its ability to achieve these strong results while significantly reducing the number of trainable parameters and training time. The paper explicitly states that FLoRA can achieve comparable performance to full fine-tuning with an 80% reduction in parameter budget.  Additionally, Table 5 provides evidence of FLoRA's efficiency in terms of both training time and GPU memory usage compared to other methods, particularly DoRA.

* **Insightful Analysis of Low-Rank Representation:** The paper goes beyond simply presenting performance results. It provides a detailed analysis of why FLoRA's low-rank representation is more effective than other methods. Figure 4, which compares the Frobenius norm and feature amplification factor of FLoRA, LoRA, and DoRA during training, offers valuable insights. This analysis reveals that while LoRA and DoRA might initially amplify task-specific features more aggressively due to their constraints on matrix patterns, FLoRA's less constrained approach allows it to capture a broader range of task-specific information, leading to higher amplification factors upon convergence. The paper also suggests a strong correlation between the Frobenius norm of the learned changes (∆W) and the ability to capture task-specific information.

* **Clear and Well-Written:** The paper is well-structured, and the concepts are explained in a clear and concise manner.  This clarity makes it easy for readers to understand the motivation behind FLoRA, its technical details, and the significance of the results.

* **Addressing a Gap in PEFT Research:** The authors clearly identify a gap in existing PEFT research, which primarily focuses on linear layers while neglecting the complexities of high-dimensional parameter spaces. FLoRA is presented as a solution to this problem, demonstrating its novelty and potential impact on the field.

**Weaknesses:**

*   **Limited Scope of LLM Fine-tuning Experiments:** The paper lacks experiments on LLM fine-tuning (e.g., LLaMA3-8B), despite PEFT methods being commonly used in these scenarios. Evaluating FLoRA's efficacy on LLMs would enhance the paper's practicality and relevance.
*   **LoRA's Applicability in Vision-Based Models:** The paper focuses on LoRA as a comparison point for FLoRA, but LoRA is not a standard fine-tuning technique for vision-based models like ConvNext and Mask-RCNN. Comparing FLoRA to more prevalent methods like last-layer fine-tuning and prompt tuning would offer a more comprehensive evaluation of its effectiveness.
*   **Missing DoRA Results on LLaVA-1.5-7B:** The paper doesn't report the performance of DoRA on LLaVA-1.5-7B fine-tuning, despite mentioning a DoRA result of 67.6, which surpasses FLoRA's performance. Including comprehensive comparisons with state-of-the-art methods like DoRA is crucial for establishing FLoRA's superiority.
*   **Compatibility with Weight-Decomposed Formulations:** The paper does not address whether FLoRA is compatible with weight-decomposed formulations like those proposed by DoRA. Exploring potential integration with other advanced PEFT techniques could reveal further benefits and limitations of FLoRA.
*   **Lack of Theoretical Grounding for the Core Space:** The paper demonstrates the existence and effectiveness of a low-rank core space empirically but lacks a theoretical framework to explain these findings. Providing a theoretical foundation for the core space's properties would strengthen the paper's claims and provide valuable insights into FLoRA's underlying mechanisms.
*   **Potential Computational Overhead:** While the paper asserts that FLoRA exhibits better parameter efficiency than LoRA for larger kernel sizes, it doesn't thoroughly analyze the computational costs in diverse settings. A more comprehensive analysis of FLoRA's computational complexity in various scenarios, especially for extremely large models or complex tasks, is recommended.
*   **Focus Beyond Convolutional Kernels:** While FLoRA demonstrates its effectiveness on convolutional kernels, it should be assessed on other high-dimensional weight matrices.  Given its positioning as a fundamental low-rank adaptation method, a broader range of experiments would validate its general applicability and effectiveness.
*   **Quantifying the Impact of Structural Integrity:** The paper could benefit from a more in-depth exploration of whether preserving structural integrity truly matters in weight adaptation for computer vision. This could involve experiments or analyses that isolate the impact of structural preservation on performance, further validating FLoRA's core principle.

**Questions:**

Already listed in the Weaknesses section.

---

> ### Author Response · Authors · 2024-11-19
> **Response to Reviewer VXNC**
>
> Dear Reviewer VXNC:
>
> We would like to first extend our sincere gratitude for your time and effort in evaluating our manuscript. Your thorough evaluation and insightful comments are greatly appreciated. We will address your questions point by point and hope to resolve your concerns effectively.
>
>
> ## **Weakness 1, Limited Scope of LLM Fine-tuning Experiments**
>
> We here report the performance of FLoRA when fine-tuning LLAMA-3-8B on commonsense reasoning tasks. As seen in the following table, FLoRA still significantly outperforms LoRA when fine-tuning LLAMA-3-8B.
>
> | Method  | # Params (%) | BoolQ | PIQA | SIQA | HellaSwag | WinoGrande | ARC-e | ARC-c | OBQA | Avg. |
> | ------- | ------------ | ----- | ---- | ---- | --------- | ---------- | ----- | ----- | ---- | ---- |
> | ChatGPT | -            | 73.1  | 85.4 | 68.5 | 78.5      | 66.1       | 89.8  | 79.9  | 74.8 | 77.0 |
> | LoRA    | 0.35         | 72.3  | 86.7 | 79.3 | 93.5      | 84.8       | 87.7  | 75.7  | 82.8 | 82.9 |
> | AdaLoRA | 0.35         | 75.1  | 86.4 | 76.7 | 75.4      | 83.3       | 90.4  | 79.1  | 85.0 | 81.4 |
> | DoRA    | 0.36         | 74.5  | 88.8 | 80.3 | 95.5      | 84.7       | 90.1  | 79.1  | 87.2 | 85.0 |
> | FLoRA   | 0.35         | 74.6  | 89.7 | 81.0 | 95.0      | 85.8       | 90.5  | 81.5  | 86.8 | 85.6 |
>
>
> ## **Weakness 2, LoRA's Applicability in Vision-Based Models**
>
> Thank you for your valuable suggestion. We here compare several CV tailored PEFT method including last-layer tuning and visual prompt-tuning [1] on the VTAB-1K benchmark, consists of 19 image classification tasks across diverse domains. The average evaluation results are presented in the table below. Clearly, compared to methods specifically designed for computer vision tasks, FLoRA continues to demonstrate significant advantages.
>
>
>
> | **Method**        | # Params (M)                 | **Avg. Acc.** |
> | ----------------- | ---------------------------- | ------------- |
> |                   | **Conventional Fine-Tuning** |               |
> | Full              | 85.8                         | 68.9          |
> |                   | **PEFT Methods**             |               |
> | Last-layer Tuning | 0.08                         | 57.6          |
> | VPT-Deep [1]      | 2.03                         | 72.0          |
> | NOAH [2]          | 1.37                         | 75.5          |
> | LoRA [3]          | 1.13                         | 76.4          |
> | SSF [4]           | 0.78                         | 75.7          |
> | Adapter-P [5]     | 0.56                         | 75.5          |
> | AdaptFormer [6]   | 0.56                         | 76.7          |
> | BitFit [7]        | 0.39                         | 65.2          |
> | FacT-TT [8]       | 0.30                         | 76.7          |
> | VPT-Shallow [1]   | 0.24                         | 67.8          |
> | Compacter [9]     | 0.15                         | 74.2          |
> | FLoRA             | 0.32                         | **77.3**      |
>
> [1] 2022, ECCV, Visual prompt tuning
>
> [2] 2022, TPAMI, Neural Prompt Search
>
> [3] 2022, ICLR, LoRA: Low-rank adaptation of large language models
>
> [4] 2022, NIPS, Scaling & shifting your features: A new baseline for efficient model tuning
>
> [5] 2021,EACL, Adapterfusion: Non-destructive task composition for transfer learning
>
> [6] 2022, NIPS, Adaptformer: Adapting vision transformers for scalable visual recognition
>
> [7] 2022, ACL, Bitfit: Simple parameter-efficient fine-tuning for transformer-based masked language-models
>
> [8] 2023, AAAI, Fact: Factor-tuning for lightweight adaptation on vision transformer.
>
> [9] 2021, NIPS, Compacter: Efficient low-rank hypercomplex adapter layers
>
>
>
> ## **Weakness 3, Missing DoRA Results on LLaVA-1.5-7B**
>
> Thank you for your valuable suggestion. We have included the results of DoRA in the main table. Additionally, we have retrained FLoRA and updated our results accordingly.
>
>
>
> ## **Weakness 4, Compatibility with Weight-Decomposed Formulations**
>
> We respect your perspective; however, we believe that the core problem FLoRA aims to solve is how to achieve efficient fine-tuning while preserving structural integrity. This is fundamentally unrelated to the issue of achieving weight-decomposed formulations, which is the primary focus of DoRA. These two approaches are addressing different challenges.
>
> Moreover, even if FLoRA is not compatible with weight-decomposed formulations, this does not imply that its design is unreasonable or its performance is suboptimal. Indeed, our experimental results demonstrate that, in the vast majority of cases, FLoRA outperforms DoRA. Specifically, in CV tasks, FLoRA’s performance significantly surpasses that of DoRA, further highlighting its effectiveness and applicability.

---

> ### Author Response · Authors · 2024-11-19
> **Response to Reviewer VXNC**
>
> ## **Weakness 5, Lack of Theoretical Grounding for the Core Space**
>
> FLoRA posits that for any dimensional parameter space, whether 2D or 4D, there exists a corresponding low-rank core space. This concept is not only intuitive but also extensively validated and applied in high-order tensor decomposition theories. For instance, CP decomposition expresses high-order tensors as a summation of low-rank tensors. Moreover, tensor nuclear norm theory establishes that for high-dimensional tensors, the optimal solution to the nuclear norm corresponds to a low-rank core tensor, which effectively captures the main characteristics of the tensor. This low-rank tensor aligns with the core space hypothesized by FLoRA.
>
> Further supporting this concept, numerous studies have provided theoretical and experimental evidence affirming the existence of core spaces. For example, foundational research [10] has theoretically proven the existence of core spaces in linear parameter spaces, and other works [11-12] have experimentally validated their presence. As for convolutional parameters, many approaches have assumed the existence of a core space within the convolutional parameter spaces [13-14]. These findings collectively strengthen the argument that a core space exists.
>
> [10] 2023, NMI, Parameter-efficient fine-tuning of large-scale pre-trained language models
>
> [11] 2021, ACL, Intrinsic dimensionality explains the effectiveness of language model fine-tuning
>
> [12] 2018, ICLR, Measuring the intrinsic dimension of objective landscapes
>
> [13] 2024, arxiv, Large Convolutional Model Tuning via Filter Subspace
>
> [14] 2021, NIPS, Image Generation using Continuous Filter Atoms
>
>
>
> ## **Weakness 6, Potential Computational Overhead**
>
> Thank you for your valuable suggestion. In fact, we have conducted extensive analyses of training costs, as presented in the Table 5 in our main text. Here, we provide the training costs for fine-tuning LLAMA3-8B. We set the batch size as 1:
>
> | Method | Training Time (hrs) | GPU Memory (GB) |
> | ------ | ------------------- | --------------- |
> | LoRA   | 22.94               | 19.73           |
> | DoRA   | 51.29               | 20.25           |
> | FLoRA  | 24.12               | 19.94           |
>
> These results consistently demonstrate the efficiency of FLoRA in terms of training.
>
>
> ## **Weakness 7, Focus Beyond Convolutional Kernels**
>
> We agree with your perspective. Indeed, considering that the weights of existing large models are at most four-dimensional tensors (e.g., convolutional layers), there are no corresponding large models available for testing on other higher-dimensional weight matrices, even if we wished to do so. Therefore, in our paper, we focused our evaluation on large models with convolutional layers and those with linear layers.
>
>
>
> ## **Weakness 8, Quantifying the Impact of Structural Integrity**
>
> We appreciate your perspective on whether preserving structural integrity truly matters in weight adaptation. In fact, LoRA’s approach of adapting a high-dimensional tensor through a two-dimensional matrix essentially disrupts its structural integrity. In contrast, FLoRA maintains the structural integrity by directly constructing a low-rank core space with the same dimensionality. Therefore, by comparing the performance of LoRA and FLoRA on convolutional layers, we can directly quantify the impact of structural integrity. We can observe that FLoRA significantly outperforms LoRA. The notable performance difference between the two methods provides strong evidence that preserving structural integrity is indeed crucial for weight adaptation in computer vision tasks. Additionally, we believe this experiment serves as the most direct evidence of the impact of structural preservation on performance.

---

> > ### Comment · Reviewer_VXNC · 2024-11-25
> >
> > Thanks for the comprehensive response. I will keep the current rating for acceptance.

---

> > > ### Author Response · Authors · 2024-11-25
> > >
> > > Dear Reviewer VXNC,
> > >
> > > We sincerely thank you for your valuable and constructive reviews, and also sincerely thank you for your feedback.
> > >
> > > Best Regards,
> > >
> > > Authors.

---

### Official Review · Reviewer_B2Z8 · 2024-11-04

**Soundness:** 3
**Presentation:** 3
**Contribution:** 3
**Rating:** 6
**Confidence:** 3

**Summary:**

This paper proposes a new generalized PEFT method, FLoRA, for N-dimension parameter space. FLoRA is based on the Tucker decomposition and uses a low-rank core tensor and N low-rank matrices to reconstruct the original parameter tensors. The introduction of the low-rank core space helps preserve the structural integrity of the original parameters space. Experiments on CV, NLP and multi-modal tasks validate the effectiveness of the proposed method over baselines like LoRA and DLoRA.

**Strengths:**

1. The generalization of LoRA on N-dimension parameter space with form of Tucker decomposition is interesting and effective.

2. The authors conduct many experiments on multiple kinds of tasks and validate the effectiveness of the proposed FLoRA method.

3. The paper is well-written and easy to follow.

**Weaknesses:**

1. In Line 183, the paper mentions that “in any convolution layer, there exists a convolution core”. Is this an assumption or supported by evidences and theories? More discussion should be added to improve the reliability of this claim.

2. How does the learned convolution core help preserve the convolution’s property of spatial locality?

3. It would be better to compare the FLoRA method with some PEFT method designed for CV tasks, such as [1].

4. Can you provide the standard deviations of the main results (Table 1 etc.)?

[1] Jie, S. and Deng, Z.-H. Fact: Factor-tuning for lightweight adaptation on vision transformer. In Proceedings of the AAAI Conference on Artificial Intelligence, volume 37, pp. 1060–1068, 2023.

**Questions:**

See weaknesses.

---

> ### Author Response · Authors · 2024-11-19
> **Response to Reviewer B2Z8**
>
> Dear Reviewer B2Z8:
>
> We would like to first extend our sincere gratitude for your time and effort in evaluating our manuscript. Your thorough evaluation and insightful comments are greatly appreciated. We will address your questions point by point and hope to resolve your concerns effectively.
>
>
>
> ## **Weakness 1 and 2, Convolution Core**
>
> FLoRA posits that for any dimensional parameter space, whether 2D or 4D, there exists a corresponding low-rank core space. This concept is not only intuitive but also extensively validated and applied in high-order tensor decomposition theories. For instance, CP decomposition expresses high-order tensors as a summation of low-rank tensors. Moreover, tensor nuclear norm theory establishes that for high-dimensional tensors, the optimal solution to the nuclear norm corresponds to a low-rank core tensor, which effectively captures the main characteristics of the tensor. This low-rank tensor aligns with the core space hypothesized by FLoRA.
>
> Further supporting this concept, numerous studies have provided theoretical and experimental evidence affirming the existence of core spaces. For example, foundational researches [1-3] has theoretically and empirically proven the existence of core spaces in linear parameter spaces. As for convolution parameters, many approaches have assumed the existence of a core space within the convolutional parameter spaces [4-5]. These findings collectively strengthen the argument that a core space exists.
>
> The learned convolution core in FLoRA helps preserve the spatial locality of convolutions because it maintains the topological structure of the original convolutional space. Unlike low-rank matrix adaptation methods (e.g., LoRA) that reshape high-dimensional convolutional kernels into two-dimensional matrices, which scatter adjacent elements and disrupt spatial relationships, FLoRA operates directly in the original tensor space. It identifies a low-rank core tensor that retains the structural integrity of the convolutional kernel. This ensures that the spatial locality, a fundamental property of convolution operations where output elements depend on localized regions of the input, is effectively preserved. Consequently, FLoRA enhances the convolution layer’s ability to process spatial information without compromising the locality inherent in its parameter space.
>
> [1] 2023, NMI, Parameter-efficient fine-tuning of large-scale pre-trained language models
>
> [2] 2021, ACL, Intrinsic dimensionality explains the effectiveness of language model fine-tuning
>
> [3] 2018, ICLR, Measuring the intrinsic dimension of objective landscapes
>
> [4] 2024, arxiv, Large Convolutional Model Tuning via Filter Subspace
>
> [5] 2021, NIPS, Image Generation using Continuous Filter Atoms
>
>
>
> ## **Weakness 3, Comparison with Other Methods**
>
> Thank you for your suggestion. Following [1], we compare several CV tailored PEFT method on the VTAB-1K benchmark, consists of 19 image classification tasks across diverse domains, categorized into three groups: **Natural**, **Specialized**, and **Structured**. The average evaluation results are presented in the table below. Clearly, compared to methods specifically designed for computer vision tasks, FLoRA continues to demonstrate significant advantages.
>
> | **Method**        | # Params (M)                 | **Avg. Acc.** |
> | ----------------- | ---------------------------- | ------------- |
> |                   | **Conventional Fine-Tuning** |               |
> | Full              | 85.8  | 68.9          |
> |    | **PEFT Methods**    |      |
> | Last-layer Tuning | 0.08 | 57.6          |
> | VPT-Deep [6]      | 2.03   | 72.0          |
> | NOAH [7]          | 1.37  | 75.5          |
> | LoRA [8]          | 1.13    | 76.4          |
> | SSF [9]           | 0.78    | 75.7          |
> | Adapter-P [10]    | 0.56    | 75.5          |
> | AdaptFormer [11]  | 0.56  | 76.7          |
> | BitFit [12]       | 0.39 | 65.2          |
> | FacT-TT [13]      | 0.30                         | 76.7          |
> | VPT-Shallow [6]   | 0.24   | 67.8          |
> | Compacter [14]    | 0.15 | 74.2          |
> | FLoRA             | 0.32  | **77.3**      |
>
> [6] 2022, ECCV, Visual prompt tuning
>
> [7] 2022, TPAMI, Neural Prompt Search
>
> [8] 2022, ICLR, LoRA: Low-rank adaptation of large language models
>
> [9] 2022, NIPS, Scaling & shifting your features: A new baseline for efficient model tuning
>
> [10] 2021,EACL, Adapterfusion: Non-destructive task composition for transfer learning
>
> [11] 2022, NIPS, Adaptformer: Adapting vision transformers for scalable visual recognition
>
> [12] 2022, ACL, Bitfit: Simple parameter-efficient fine-tuning for transformer-based masked language-models
>
> [13] 2023, AAAI, Fact: Factor-tuning for lightweight adaptation on vision transformer.
>
> [14] 2021, NIPS, Compacter: Efficient low-rank hypercomplex adapter layers
>
> ## **Weakness 4, Standard Deviations**
>
> Thank you for your suggestions. We have added standard deviations of the results in our manuscript.

---

### Official Review · Reviewer_UDNJ · 2024-11-10

**Soundness:** 3
**Presentation:** 3
**Contribution:** 2
**Rating:** 6
**Confidence:** 4

**Summary:**

This paper introduces FLoRA, a parameter-efficient fine-tuning (PEFT) method for adapting pre-trained foundation models across various parameter spaces while preserving the model’s structural integrity. FLoRA minimizes the computational burden of full fine-tuning by maintaining the original parameter organization, allowing efficient task-specific adaptability of different architectures. This approach supports scalable adaptation in large models, optimizing both resource use and performance. Empirical and theoretical evaluations show FLoRA’s robustness, highlighting its promise as an efficient approach for model adaptation.

**Strengths:**

+ The motivation behind maintaining the topological structure of the pre-trained matrices is compelling, providing a strong foundation for the proposed approach.

+ The paper is well-organized and easy to follow, with each section clearly building upon the previous.

+ The analysis section aligns well with the proposed objectives, and the approach to evaluating whether the core space is low-rank adds depth to the analysis.

**Weaknesses:**

- The work appears to have similarities with existing approaches such as LoTR and LoKR; however, the paper does not adequately address how FLoRA distinguishes itself from these methods. This lack of clarification raises serious concerns about the novelty of the proposed approach.
- The manuscript does not provide sufficient evidence to demonstrate how the representations in FLoRA surpass those of LoRA and DoRA. It remains unclear how FLoRA broadens parameter adjustments and how this relates to enhancing parameter learning flexibility.
- The feature amplification factor introduced in the study may lack a direct correlation with actual task performance gains, making it difficult to interpret its significance in relation to improved task-specific outcomes. Currently, the correlation is indicated only using a single dataset and model architecture.
- The correlation observed between the Frobenius norm and feature amplification could be incidental rather than causal. There is insufficient evidence to support the claim that a larger norm consistently leads to enhanced task-specific performance.
- The scalability and computational efficiency of the proposed approach may be limited when applied to extremely high-dimensional tensors, potentially affecting its practicality for very large foundation models.

**Questions:**

Please see weaknesses above. Below are additional questions:
* What exactly is the difference between the current method and existing methods, particularly LoTR? The decomposition appears identical since both methods use Tucker decomposition, yet the paper claims that there are differences. It is hard to appreciate the paper without knowing about the differences between this work and past methods.
* How is the feature amplification factor considered a reliable indicator of task-specific information amplification, and what insights does it provide regarding task-specific performance? Furthermore, what methods are employed to measure task-specific information in the context of this study?
* What significance does the average Frobenius norm of $\Delta W$ hold for FLoRA's performance during fine-tuning? Can the magnitude of this norm be directly correlated to the effectiveness of task-specific adaptation? The presented correlation is based solely on a single sample, and there is no clear explanation provided for why this correlation would exist in the first place.
* How does the feature amplification factor of FLoRA in the DeBERTaV3-base model compare to that of other fine-tuning methods applied to the CoLA dataset? Does FLoRA exhibit a distinct pattern in terms of norm growth or amplification factor throughout the training process?
* If different versions of the DeBERTa model were evaluated, would the Frobenius norm or amplification factor be affected by changes in model architecture, or are these metrics consistently reliable across various transformer-based models?
* Without a direct comparison of the Frobenius norm and amplification factor metrics against alternative fine-tuning approaches, such as standard LoRA or full fine-tuning, it is challenging to assert FLoRA's superiority. If these metrics do not demonstrate a clear advantage over other methods, this could weaken the strength of the conclusions drawn in the paper.

---

> ### Author Response · Authors · 2024-11-19
> **Response to Reviewer UDNJ**
>
> Dear Reviewer UDNJ:
>
> We would like to first extend our sincere gratitude for your time and effort in evaluating our manuscript. Your thorough evaluation and insightful comments are greatly appreciated. We will address your questions point by point and hope to resolve your concerns effectively.
>
> ## **Weakness 1, Question 1, Comparison with LoTR**
> **We have already compared the differences between FLoRA and LoTR in the related work.** For your convenience, we have pasted the relevant content here:
>
> There are some works like LoTR also adopt low-rank tensor adaptations, but they are unrelated to FLoRA.
> 1. Usage and Impact. Based on low-rank tensor adaptation, FLoRA directly models $\Delta\mathbf{W}$ for the weight matrix of each layer, maintaining its spatial structure. However, LoTR concatenates $\Delta\mathbf{W}$ across all layers and then applies low-rank tensor adaptation, assuming all layers share same weight matrices, which is a strong assumption that leads to suboptimal performance. Besides, LoTR disrupts the original parameter space structure although they use low-rank tensor adaptation.
> 2. Motivation and Details. FLoRA aims to model $\Delta\mathbf{W}$ while preserving the N-dimensional topological structure, meanwhile addressing the limitations of other methods focusing only on linear weights. However, LoTR aims to achieve extreme parameter efficiency, and is tailored for linear weights, limiting its applications.
>
> Moreover, **using the same decomposition form in PEFT is indeed very common.** Many works like AdaLoRA, RoSA, TriLoRA, PISSA and MiLoRA use SVD, but they hold quite distinct objectives and implementations.
>
> We also provide the average performance of FLoRA and LoTR on NLU using RoBERTa-base in the following table. It is evident that under similar parameter budgets (0.32M), FLoRA significantly outperforms LoTR.
> |Fully FT|LoTR|FLoRA|
> |-|-|-|
> |86.4|79.0|86.2|
> ## **Weakness 2, Representations Superiority of FLoRA**
> Let $\\mathbf{W}^\*\\in\\mathbb{R}^{n\\times m}$ be the optimal weight for a specific linear layer with frozen weight $\\mathbf{W}$, and $\\Delta\\mathbf{W}^{\*}=\\mathbf{W}^{\*}-\\mathbf{W}$ is the optimal change. Generally, the low-rank representation of LoRA’s series can be represented as $\\Delta\\mathbf{W}=\\mathbf{AGB}$, where $\\mathbf{A}\\in\\mathbb{R}^{n\\times r}$, $\mathbf{B}\in\mathbb{R}^{r\times m}$, $\mathbf{G}\in\mathbb{R}^{r\times r}$ and $r\ll\{n,m\}$, and the essential difference of these methods is the matrix pattern of $\\mathbf{G}$. In FLoRA, $\\mathbf{G}$ is arbitrary, in AdaLoRA, it is diagonal, and identity in LoRA.
>
> Ideally, we have $\\mathbf{G}=\\mathbf{A}^{\\dagger}\\Delta\\mathbf{W}^{\*}\\mathbf{B}^{\\dagger}$, where $\\mathbf{A}^{\\dagger}$ and $\\mathbf{B}^{\\dagger}$ are the Moore-Penrose Pseudo-inverses of $\\mathbf{A}$ and $\\mathbf{B}$, respectively. If $\\mathbf{G}$ is an arbitrary matrix such as FLoRA, it is obvious that no constraints are imposed on $\\mathbf{A}$ and $\\mathbf{B}$, since for any choice of $\\mathbf{A}$, $\\mathbf{B}$, we can always find a $\\mathbf{G}$. Considering when $\\mathbf{G}$ is diagonal or identity, we use SVD on $\\Delta\\mathbf{W}^{\*}=\\mathbf{U}\\mathbf{\\Sigma}\\mathbf{V}^\\mathsf{T}$ and $\\mathbf{G}=\\mathbf{A}^{\\dagger}\\mathbf{U}\\mathbf{\\Sigma}\\mathbf{V}^\\mathsf{T}\\mathbf{B}^{\\dagger}$. We then require $\\mathbf{A}^{\\dagger}=\\mathbf{D}\_1\\mathbf{U}^\\mathsf{T}$ and $\\mathbf{B}^{\\dagger}=\\mathbf{V}\\mathbf{D}\_2$, where $\\mathbf{D}\_1$ and $\\mathbf{D}\_2$ are rectangular diagonal matrices. It means $\\mathbf{G}=\\mathbf{D}\_1\\mathbf{\\Sigma}\\mathbf{D}\_2$. If $\\mathbf{G}$ is diagonal like AdaLoRA, we can constrain $\\mathbf{D}\_1=\\mathbf{I}\_{r\\times n}$ and leave $\\mathbf{D}\_2$ unrestricted, since any $\\mathbf{D}_2$ can correspond to a $\\mathbf{G}$. However, if $\\mathbf{G}$ is identity such as LoRA, and with $\\mathbf{D}\_1=\\mathbf{I}\_{r\\times n}$,  $\\mathbf{D}\_2$ must be $(\\mathbf{D}\_1\\mathbf{\\Sigma})^\\dagger$.
>
> Thus, to achieve the optimal change $\\Delta\\mathbf{W}^\*$, FLoRA imposes no constraints on $\\mathbf{A}$, $\\mathbf{B}$, AdaLoRA has constraints on $\\mathbf{A}$, and LoRA further restricts $\\mathbf{A}$ and $\\mathbf{B}$. However, these constraints are not applied during training, which means that LoRA and AdaLoRA are less likely to learn the optimal change compared to FLoRA. Further, even if specific matrix patterns are constrained during training, FLoRA’s greater degree of freedom and stronger expressive power make it much easier to achieve better results [1,2].
>
> Moreover, some works have also demonstrated the effectiveness of FLoRA [3,4].
>
> [1] 2023 ICLR, AdaLoRA: Adaptive Budget Allocation for Parameter-Efficient Fine-Tuning
>
> [2] 2024 KNOWL INF SYST, Model Complexity of Deep Learning: A Survey
>
> [3] 2024, arXiv, See Further for Parameter Efficient Fine-tuning by Standing on the Shoulders of Decomposition
>
> [4] 2024, EMNLP, Mixture-of-Subspaces in Low-Rank Adaptation

---

> ### Author Response · Authors · 2024-11-19
> **Response to Reviewer UDNJ**
>
> ## **Weakness 3, 4, Question 2, 3, 4, 5, 6, Feature Amplification Factor and Frobenius Norm**
>
> **We have carefully reviewed and organized the feedback from Reviewer UDNJ and identified that the primary concerns center around the Feature Amplification Factor and Frobenius Norm, comprising 2 weaknesses and 5 questions out of a total of 11 comments.** These concerns are directly related to the experimental analysis section, specifically in Section 5.2, Lines 415 to 431 of the main text. Below, we provide detailed responses and hope these address Reviewer UDNJ’s concerns comprehensively.
>
>
>
> ### **What is feature amplification factor? (Question 2)**
>
> The **Feature Amplification Factor** is a metric introduced by LoRA to measure how much task-specific information is amplified by $\\Delta \\mathbf{W}$. Simply put, it calculates the ratio of the task-specific information learned by the model (represented by the Frobenius norm of $\\Delta \\mathbf{W}$) to the amount of information obtained by projecting the pre-trained weights $\\mathbf{W}$ onto the task-specific directions. Mathematically, the Feature Amplification Factor is defined as $\\frac{\\|\\Delta\\mathbf{W}\\|\_F}{\\| \\mathbf{U}^\\mathsf{T}\\mathbf{W}\\mathbf{V}\\|\_F}$, where $\\mathbf{U}$ and $\\mathbf{V}$ are the top $r$ left and right singular vectors of $\\Delta\\mathbf{W}$. Here, $r$ is the rank configuration of PEFT methods like LoRA.
>
> This metric provides insight into how effectively the model adapts to downstream tasks by quantifying the relative contribution of the learned task-specific updates, and it has been widely adopted in PEFT works to evaluate the amount of task-specific information learned by the model [5-7]. If the Reviewer UDNJ has further questions regarding the definition and calculation or others of the Feature Amplification Factor, we recommend referring to Section 7 of the original LoRA paper for additional details. We also warmly welcome Reviewer UDNJ to discuss these aspects with us further.
>
> [5] 2024, arXiv, MiLoRA: Harnessing Minor Singular Components for Parameter-Efficient LLM Fine-tuning
>
> [6] 2024, arXiv, Unleashing the Power of Task-Specific Directions in Parameter Efficient Fine-tuning
>
> [7] 2024, arXiv, Semantic are Beacons: A Semantic Perspective for Unveiling Parameter-Efficient Fine-Tuning in Knowledge Learning
>
>
>
> ### **What is the relationship between feature amplification factor and model performance? (Weakness 3, Question 2)**
>
> Before addressing this question, we would like to clarify that we have never stated in our paper that the Feature Amplification Factor or Frobenius Norm is directly related to task performance. We have never suggested that a higher Feature Amplification Factor directly translates to better task performance. What we consistently emphasize is that a larger Feature Amplification Factor or Frobenius Norm in FLoRA signifies that it contains more task-specific information.
>
> Indeed,  although LoRA does not explicitly state a relationship between these factors and task performance, its experimental results show a clear correlation with amplification levels. Moreover, many existing works [5-7] implicitly or explicitly suggest that a larger Feature Amplification Factor correlates with improved model performance. For instance, [6-7] directly highlights that amplifying more task-specific information can enhance model performance.
>
> Therefore, while we did not explicitly state in our paper that there is a necessary link between the two, based on our review of existing literature and as part of our discussion with Reviewer UDNJ, we believe there is evidence to suggest a correlation: that is, larger amplification factors are often associated with better model performance.

---

> ### Author Response · Authors · 2024-11-19
> **Response to Reviewer UDNJ**
>
> ### **What is the relationship between feature amplification factor and Frobenius norm of $\\Delta\\mathbf{W}$? (Weakness 4, Question 3)**
>
> In our experiments, we observed that the final value of the Frobenius norm of $\\Delta \\mathbf{W}$ is positively correlated with the Feature Amplification Factor, meaning that as one increases, the other also tends to increase. This was somewhat unexpected but also intuitively reasonable: given that the Feature Amplification Factor is directly calculated using the Frobenius norm of $\\Delta \\mathbf{W}$, it is logical to assume some intrinsic connection between the two. At the time of submission, we had not fully understood why this alignment occurred. However, with further analysis, we can now offer an explanation for this phenomenon:
>
> Firstly, the **Feature Amplification Factor** is defined as the ratio of the Frobenius norm of $\\Delta \\mathbf{W}$, i.e., $\\|\\Delta\\mathbf{W}\\|\_F$, to the amount of information in the pre-trained weights $\\mathbf{W}$ projected onto the task-specific directions, denoted as $\\|\\mathbf{U}^\mathsf{T}\\mathbf{W}\\mathbf{V}\\|\_F$. Since the pre-trained weight matrix $\mathbf{W}$ is frozen during training, the value of $\\|\\mathbf{U}^\\mathsf{T}\\mathbf{W}\\mathbf{V}\\|\_F$ depends solely on $\\mathbf{U}$ and $\\mathbf{V}$, which are two orthonormal matrices derived from $\\Delta \\mathbf{W}$ and represent task-specific directions.
>
> In [6], the authors highlight that at any step during training, $\\Delta \\mathbf{W}$ effectively captures the task-specific directions. Consequently, from the beginning to the end of training, the changes in $\\mathbf{U}$ and $\\mathbf{V}$ are minimal (as also validated in [6]), which implies that $\\|\\mathbf{U}^\\mathsf{T}\\mathbf{W}\\mathbf{V}\\|\_F$  remains nearly constant. In other words, a larger $\\|\\Delta\\mathbf{W}\\|\_F$ does indeed correspond to a larger Feature Amplification Factor. Therefore, while we fully respect Reviewer UDNJ’s perspective that “Frobenius norm and Feature Amplification could be incidental rather than causal,” based on the above analysis, we believe that their relationship is inherent and inevitable.
>
> Building on our earlier analysis of the relationship between amplification factors and model performance, it is evident that a larger Frobenius norm of $\\Delta\\mathbf{W}$ corresponds to better performance. This is a surprising yet logical and exciting conclusion. In the paper, we hypothesized that larger Frobenius norm values of $\\Delta\\mathbf{W}$ allow for encoding more task-specific information, thereby effectively amplifying the task-specific signals in the frozen weights. Upon further reflection, this insight aligns with the findings of [6] but from a different perspective: amplifying task-specific information leads to better task performance. Moreover, directly increasing the Frobenius norm of $\\Delta\\mathbf{W}$ emerges as a more straightforward approach to achieve this amplification.
>
> We sincerely appreciate the opportunity provided by the Reviewer UDNJ to explore the underlying reasons behind these phenomena together. In fact, uncovering the essence behind counterintuitive observations is the true essence of research. We  look forward to further discussions and analyses of these intriguing phenomena with Reviewer UDNJ.
>
>
>
> ### **Is this phenomenon universal? (Weakness 3, Question 3, 5)**
>
> We used one model and one dataset in our experiments to illustrate the phenomenon and showcase our findings. Unfortunately, it is nearly impossible to theoretically prove that this phenomenon is universal. However, in our broader experiments, we observed similar trends across different datasets and models. We have updated our manuscript, and add Figure 7 in Appendix. Here we fine-tune DeBERTaV3-large on SST-2 dataset, and the results align well with our aforementioned analysis.
>
> Furthermore, to avoid randomness, we did not selectively choose $\\Delta \\mathbf{W}$ from a specific layer of the model. Instead, we computed the mean of  $\\Delta \\mathbf{W}$ across all layers. We believe this averaging approach better reflects the overall trend, providing a more comprehensive and robust representation of the observed phenomenon.
>
> Therefore,  we believe this phenomenon can be consistently observed across the experiments we conducted.

---

> ### Author Response · Authors · 2024-11-19
> **Response to Reviewer UDNJ**
>
> ### **FLoRA’s learning pattern and comparison with other fine-tuning methods (Question 4, 6)**
>
> In Figure 4, we have already compared the Frobenius norm and amplification factor metrics with other methods, such as LoRA and DoRA, and presented the trends exhibited by different methods.
>
> It can be observed that during the training process, LoRA and DoRA amplify more task-specific features than FLoRA in the early stages. As discussed earlier, these two methods impose strong constraints on matrix patterns, resulting in a distinct directional learning pattern at the beginning, which contributes to their larger initial values. However, such constraints on matrix patterns may not always be optimal for downstream tasks, as downstream datasets often possess diverse properties. Consequently, their amplification factors upon convergence are smaller than that of FLoRA.
>
> Therefore, FLoRA ultimately amplifies more task-specific information, which further demonstrates its superior expressive capacity.
>
>
>
> Indeed, **both Reviewer VXNC and Reviewer WIU6 acknowledged our exploration of the feature amplification factor and Frobenius norm as an** ***insightful analysis of low-rank representation***. We hope that the above response could address Reviewer UDNJ’s concerns effectively.
>
>
>
> ## **Weakness 5,  Scalability and Computational Efficiency**
>
> We have conducted extensive analyses of scalability and computational efficiency, as presented in the Table 5 in our main text. These results consistently demonstrate the efficiency of FLoRA in terms of training.
>
> Indeed, considering that the weights of existing prevailing large foundation models are at most four-dimensional tensors (e.g., convolutional layers), there are no corresponding large models available for testing on other extremely higher-dimensional weight matrices, even if we wished to do so. Therefore, in our paper, we focused our evaluation on large models with convolutional layers and those with linear layers.

---

> > ### Comment · Reviewer_UDNJ · 2024-12-01
> > **Response to authors' rebuttal**
> >
> > I thank the authors for the detailed responses. The comments address some of my queries, below are a few follow-up questions. I'd appreciate the authors' thoughts on them:
> >
> > * W1: Would it be fair to say that the primary difference between LoTR and FLoRA is that layerwise decomposition (but using the same decomposition strategy otherwise)? Would you have any deeper insights as to why this change results in a significant boost in performance?
> > * W3: I have a followup question on this. In the ideal case, $||U^TWV|| = ||W||$ (See https://nla.skoltech.ru/archive/2018/lectures/html/04%202%20Matrix%20norms%20and%20unitary%20matrices.html for a sample reference). This is a constant, and hence the denominator of the feature amplification factor expression is constant in this case. This would mean that the feature amplification factor is directly proportional to the the numerator i.e $||\Delta W||_F$. This term may not directly be related to the performance -- for example, training with L2 regularization would decrease the value, but may improve the performance. This issue makes it difficult to conclude that such a relationship is inevitable.

---

> > > ### Author Response · Authors · 2024-12-01
> > >
> > > Dear Reviewer UDNJ,
> > >
> > > We sincerely thank you for your response.
> > >
> > > ## LoTR and FLoRA
> > >
> > > From the perspective of the final matrix representation, the primary difference between LoTR and FLoRA indeed lies in the layerwise decomposition. However, this distinction arises from a series of strong assumptions made by LoTR. Specifically, LoTR applies Tucker decomposition-based adaptation to the weight deltas ($\\Delta \\mathcal{W}$) across all layers (treated as 3D tensors) in the form of $\\mathcal{G} \\times_1 \\mathbf{A} \\times_2 \\mathbf{B} \\times_3 \\mathbf{C}$ , and further assumes that $\\mathbf{C}$ is an identity matrix $\\mathbf{I}$. This assumption simplifies the representation to a form similar to FLoRA’s, namely $\\mathbf{A} \\mathbf{G} \\mathbf{B}$.
> > >
> > > It is clear that this process involves two strong assumptions:
> > >
> > > ​	1.	$\\mathbf{C}$ is an identity matrix.
> > >
> > > ​	2.	All layers share the same core matrix $\\mathbf{G}$.
> > >
> > > These are evidently strong constraints that may not hold in many practical scenarios. As a result, LoTR’s performance significantly lags behind FLoRA in a variety of tasks, as the rigid assumptions limit its flexibility and adaptability to diverse settings.
> > >
> > >
> > >
> > > ## Dominator of the feature amplification factor
> > >
> > > Thanks for the reviewer's insightful comment. There may be some misunderstanding on the denominator of the feature amplification factor.
> > >
> > > We first want to point out that for a weight matrix $\\mathbf{W}\\in\\mathbb{R}^{n\\times m}$, $\\mathbf{U}\\in\\mathbb{R}^{n\\times r}$ and $\\mathbf{V}\\in\\mathbb{R}^{m\\times r}$, the top $r$ singular vectors of $\\Delta\\mathbf{W}$, are not square matrices. For $\\|\\mathbf{U}^\\mathsf{T}\\mathbf{W}\\mathbf{V}\\|\_F^2$, we have $\\|\\mathbf{U}^\\mathsf{T}\\mathbf{W}\\mathbf{V}\\|\_F^2 = tr(\\mathbf{V}^\\mathsf{T}\\mathbf{W}^\\mathsf{T}\\mathbf{U}\\mathbf{U}^\\mathsf{T}\\mathbf{W}\\mathbf{V})$. If $\\mathbf{U}$ and $\\mathbf{V}$ are square matrices, we have $\\mathbf{U}\\mathbf{U}^\\mathsf{T}=\\mathbf{I}$ and $\\mathbf{V}\\mathbf{V}^\\mathsf{T}=\\mathbf{I}$, therefore $\\|\\mathbf{U}^\\mathsf{T}\\mathbf{W}\\mathbf{V}\\|\_F^2 =tr(\\mathbf{W}^\\mathsf{T}\\mathbf{W})$. However, since $r<n,m$, $\\mathbf{U}\\mathbf{U}^\\mathsf{T}\\neq\\mathbf{I}$ and $\\mathbf{V}\\mathbf{V}^\\mathsf{T}\\neq\\mathbf{I}$, therefore the denominator of the feature amplification factor expression **is not the constant** through the whole training phase. In this case, the feature amplification factor may directly be related to the performance.

---

> > > > ### Comment · Reviewer_UDNJ · 2024-12-01
> > > >
> > > > Thank you for the clarifications, I have updated my score accordingly.

---

> > > > > ### Author Response · Authors · 2024-12-01
> > > > >
> > > > > Dear Reviewer UDNJ,
> > > > >
> > > > > We sincerely thank you for your time and effort in evaluating our manuscript and also your valuable reviews.
> > > > >
> > > > > Best Regards,
> > > > >
> > > > > Authors.

---

### Author Response · Authors · 2024-11-19
**Common Response**

Dear AC and all reviewers:

We sincerely thank you for the time and effort you have dedicated to reviewing our work. We deeply appreciate the valuable feedback provided by the reviewers, which has allowed us to further improve the quality of our work. In response to the reviewers’ constructive comments, we have made the following revisions to our paper:

1. In response to Reviewer UDNJ weakness 2 and Reviewer wiu6 weakness 1, we have added a detailed theoretical analysis of the effectiveness of FLoRA’s low-rank representation in Appendix B.1.

2. In response to Reviewer UDNJ, we have included a deeper discussion on the Feature Amplification Factor and the Frobenius norm of $\Delta \mathbf{W}$ in Appendix B.2.
3. In response to Reviewer B2Z8 weakness 3 and Reviewer VXNC weakness 2, we have added experimental results on the VTAB-1k benchmark and compared them with CV-tailored methods, as shown in Appendix A.3.

4. In response to Reviewer B2Z8 weakness 4, we have included the standard deviations of the results presented in Table 3 to provide a more comprehensive statistical analysis.
5. In response to Reviewer VXNC weakness 1and Reviewer wiu6 weakness 1, we have conducted additional experiments using LLaMA3-8B on commonsense reasoning tasks, with the results included in Appendix A.2.
6. In response to Reviewer weakness 3, we have supplemented the experimental results for DoRA on the LLaVA benchmark, as presented in Table 4.

We deeply value each review provided by the reviewers and sincerely appreciate how these comments have helped us enhance the quality of our paper. We hope these revisions address the concerns raised by the reviewers. Thank you again for your insightful feedback and guidance.

Sincerely,

Authors

---

### Author Response · Authors · 2024-11-22

Dear Reviewers,

May we kindly ask if our responses have addressed your concerns? We look forward to further discussions and feedback from you!

Sincerely,

Authors

---

### Meta-Review · Area_Chair_HFyx · 2024-12-18

**Metareview:**

The paper presents a compelling advancement in PEFT methods through FLoRA, which effectively addresses the challenges of adapting high-dimensional parameter spaces. It provides strong empirical evidence of significant performance gains over existing methods like LoRA and DoRA across diverse tasks, including computer vision, NLP, and multimodal applications, while maintaining parameter efficiency and practicality. During the rebuttal phase, the authors were able to sufficiently address the concerns from the reviewers. Overall, the paper has sufficient contributions and potential impact on the community.

**Additional Comments On Reviewer Discussion:**

N/A

---

### Decision · Program_Chairs · 2025-01-22

Accept (Poster)